# Activation of the CA2-ventral CA1 pathway reverses social discrimination dysfunction in *Shank3B* knockout mice

Elise C. Cope[1,2], Samantha H. Wang[1], Renée C. Waters[1], Isha R. Gore[1], Betsy Vasquez[1], Blake J. Laham[1] & Elizabeth Gould [1]✉

Mutation or deletion of the *SHANK3* gene, which encodes a synaptic scaffolding protein, is linked to autism spectrum disorder and Phelan-McDermid syndrome, conditions associated with social memory impairments. *Shank3B* knockout mice also exhibit social memory deficits. The CA2 region of the hippocampus integrates numerous inputs and sends a major output to the ventral CA1 (vCA1). Despite finding few differences in excitatory afferents to the CA2 in *Shank3B* knockout mice, we found that activation of CA2 neurons as well as the CA2-vCA1 pathway restored social recognition function to wildtype levels. vCA1 neuronal oscillations have been linked to social memory, but we observed no differences in these measures between wildtype and *Shank3B* knockout mice. However, activation of the CA2 enhanced vCA1 theta power in *Shank3B* knockout mice, concurrent with behavioral improvements. These findings suggest that stimulating adult circuitry in a mouse model with neurodevelopmental impairments can invoke latent social memory function.

Social memory is an important ability that gives rise to adaptive social interactions. Social memory dysfunction has been reported in several neuropsychiatric disorders, including autism spectrum disorder (ASD), schizophrenia, and major depressive disorder[1–5]. Deficits in the ability to recognize individuals and to make associations between individuals and specific traits, emotions, events, and settings can significantly impair the formation and maintenance of social relationships[6,7]. These findings raise the importance of identifying the mechanisms of social memory as potential targets for therapeutic intervention.

Evidence from human studies indicates hippocampal involvement in social memory[8,9], including reports of hippocampal abnormalities in individuals with conditions associated with social memory dysfunction[10–13]. A growing number of studies aim to investigate mechanisms of social memory using experimental animals, with many reporting that the hippocampus plays an important role in this function[14–16]. Circuitry supporting social memory has been identified in the rodent hippocampus, with studies describing the CA2 as a social memory "hub" that integrates signals from a variety of afferents[17]. Afferents to the CA2 are both extrahippocampal, from the

supramammillary nucleus and paraventricular nucleus of the hypothalamus, the cholinergic basal forebrain, and the entorhinal cortex, and intrahippocampal, from the dentate gyrus and CA3 region[18–22]. Studies have shown that afferents from the lateral entorhinal cortex, hypothalamus, and cholinergic basal forebrain to the CA2 play important roles in social novelty recognition and social discrimination[19–23].

Several types of network oscillatory patterns in the hippocampus have been associated with social behavior. Recent studies have shown that social stimuli result in changes to sharp wave ripples (SWRs), high frequency oscillatory events known to be associated with nonsocial memory consolidation and retrieval[24,25], in both the CA2[26] and ventral CA1[27,28] (vCA1). The CA2 communicates with the vCA1[29,30], which serves as the main hippocampal output carrying social memory information[31,32]. Social stimuli also increase CA1 oscillations in the gamma and theta ranges[30,33], and several animal models of social dysfunction exhibit abnormal hippocampal gamma and theta power[34–36]. Taken together, these findings suggest that atypical SWRs, as well as gamma and theta rhythms, in the CA2-vCA1 pathway may contribute to social memory impairment.

[1]Princeton Neuroscience Institute, Princeton University, Princeton, NJ 08544, USA. [2]Present address: Department of Neuroscience, University of Virginia School of Medicine, Charlottesville, VA 22908, USA. ✉e-mail: goulde@princeton.edu

ASD represents a range of conditions with defining core symptoms (social impairments and restrictive interests/repetitive behaviors) of differing severity, as well as a wide range of potential comorbidities (intellectual disability, anxiety disorders, epilepsy)[37]. Perhaps not surprisingly, given the wide range of symptom presentation, the etiology of ASD appears to be multifactorial[38,39]. It is generally accepted that ASD arises through a complex interaction between genes and the environment[38]. Genome-wide association studies have identified over a hundred genes linked to ASD, with a significant number playing a role in neuronal communication[40]. Due to this multifactorial etiology with clustered risk genes, understanding the mechanisms of social memory and associated dysfunctional neuronal circuitry may reveal translatable discoveries beyond genetic approaches. Among the genes shown to be linked to ASD is *SHANK3*, whose mutation or deletion also causes Phelan-McDermid syndrome, a condition that often presents with similar symptoms to ASD[41,42]. The *SHANK3* gene encodes the SH3 and multiple ankyrin repeat domains 3 protein, a synaptic scaffolding protein that functions to anchor receptors and ion channels to the postsynaptic site[43]. Shank3 protein also plays a role in excitatory synapse and dendritic spine formation[44]. While *SHANK3* mutations are only strongly linked to a subset of ASD cases, the broader way that altered synaptic transmission and neuronal communication produce ASD symptomology may also provide translatable insight. Several *Shank3* models have been created in nonhuman primates and rodents using transgenic and CRISPR technologies. These models, which involve mutations or knockouts of parts or all of the *Shank3* gene, exhibit behavioral phenotypes analogous to the core symptoms of ASD, including social communication deficits and excessive repetitive behaviors[45–49]. *Shank3* knockout (KO) mice have also been shown to exhibit social recognition deficits[28,50], which may be analogous to deficits in recognizing familiar faces and familiar voices reported in some people with ASD and Phelan-McDermid syndrome[1,4,51–55]. Taken together, these findings suggest that studies using *Shank3* animal models have the potential to provide translationally relevant information about circuits impacted in ASD and, importantly, to suggest points of intervention for improving function.

In the majority of cases, ASD symptoms persist throughout life[56]. Studies have shown that adults with ASD experience lower life satisfaction[57–60] and are more likely to be unemployed and socially isolated than peers without ASD, with social impairments and socially disruptive behaviors being significant predictors[57,58,61–63]. These findings emphasize the need to identify interventions for adults with ASD. Along these lines, recent studies have shown that deep brain stimulation of the striatum diminishes excessive repetitive behavior and improves social communication in adults with ASD[64,65]. These promising results raise the possibility that activating brain regions involved in social memory might restore this ability as well. To investigate this in a mouse model of social dysfunction, we first confirmed that *Shank3B* KO mice have deficient social discrimination abilities, in that they respond similarly to novel and familiar mice, despite the fact that most excitatory afferents to the CA2 appear to be similar to their wildtype (WT) littermates, including those that have been directly linked to social novelty recognition and social discrimination. We next used chemogenetics to activate excitatory neurons in the CA2 region, and also more directly the CA2-vCA1 pathway, both of which restored social discrimination abilities in adult *Shank3B* KO mice. We found that *Shank3B* KO mice have typical oscillatory rhythms in the SWR, gamma, and theta ranges in the vCA1, yet DREADD-induced improvement of social discrimination was accompanied by a boost in theta power.

## Results

### *Shank3B* KO mice have impaired social discrimination, but intact object location memory

We first confirmed previous reports that *Shank3B* KO mice have low social investigation times as well as impaired social discrimination memory[28,50]. We utilized a three-trial direct social interaction test, with each trial separated by 24 h, in which mice were exposed to a novel mouse (Novel 1) in trial 1, re-exposed to the same mouse (Familiar) in trial 2, and exposed to a second novel mouse (Novel 2) in trial 3 (Fig. 1a). Adult WT mice typically prefer novelty and thus investigate novel mice more than familiar mice. To assess whether there was a sex difference related to genotype, we carried out a three-way ANOVA (sex × genotype × trial) and found no significant interaction between sex and genotype ($F$ (1, 21) = 0.1275; $p$ = 0.7246), so male and female data were collapsed for all subsequent analyses. We found the WT mice to have significantly higher interaction times for the novel 1 and novel 2 than for the familiar mouse (Fig. 1b, Table S1). *Shank3B* KO mice did not have significant differences in interaction times across trials (Fig. 1b, Table S1) and had difference scores of significantly lower magnitudes that were approaching zero (Fig. 1c, Table S1), indicating lower preference or discrimination of novelty. *Shank3B* KO mice had low novel interaction times as compared to WT, perhaps indicating reduced novelty detection (Fig. 1b, Table S1). It should be noted that *Shank3B* KO mice also exhibit a hypomobility phenotype[48] that may contribute to the reduced novel investigation effect. We then examined whether this difference in investigation time generalized to other hippocampal-dependent cognitive tests, particularly without social components. Using the object location memory test (Fig. 1d), which requires the hippocampus[66], we found that *Shank3B* KO mice were similarly capable as WT mice at distinguishing between objects in a novel versus familiar location (Fig. 1e, Table S1). There was also no difference between genotypes in their total investigation times of the objects (Fig. 1f, Table S1) or time to reach the criterion for familiarization (Fig. 1g, Table S1). Taken together, these data show that *Shank3B* KO mice have impairments in social discrimination abilities, yet have other intact hippocampal processes, including non-social novelty detection.

### *Shank3B* KO mice exhibit similar avoidance behavior to WT mice

To consider the possibility that reduced social interaction of the novel stimulus mouse in *Shank3B* KO mice is the result of high levels of general avoidance behavior, we examined behavior on the elevated plus maze (EPM) (Fig. S1A, Table S2). Indicators of avoidance behavior are lower time and fewer entries onto open arms. *Shank3B* KO mice showed no differences on these measures (Fig. S1B, C, Table S2), which suggests that *Shank3B* KO mice do not display more avoidance behavior on the EPM relative to WT mice, which is consistent with some[48] but not all[67] previous findings. *Shank3B* KO mice did show significant differences with lower percentage of time spent in the center (Fig. S1B, Table S2) and fewer entries into the closed arms, as compared to WT mice (Fig. S1C, Table S2). These differences might be better explained by activity or decision-making differences rather than avoidance behavior.

### Most afferents to CA2 appear similar in WT and *Shank3B* KO mice

Shank3 is a synaptic scaffolding protein that participates in the formation of excitatory synapses throughout the brain, and its disruption has been shown to reduce synapses and overall connectivity[68,69]. To investigate whether *Shank3B* KO mice exhibit differences in CA2 inputs, we examined afferent populations that have been linked to social discrimination or novelty recognition. First, we examined adult-born granule cells (abGCs), which are known to project to the CA2[70] and have been linked to social memory[71]. We analyzed the intensity of 3R-Tau, a microtubule-associated protein that labels abGC cell bodies and their mossy fibers[72], in the CA2. Compared to WT mice, we found that *Shank3B* KO mice have lower 3R-Tau+ mossy fiber intensity in the CA2 (Figs. 1h, S3B, S4A, Table S1). Since fewer afferents from abGCs to the CA2 might result from lower numbers of abGCs in the dentate gyrus of *Shank3B* KO mice, we then examined the number of 3R-Tau+

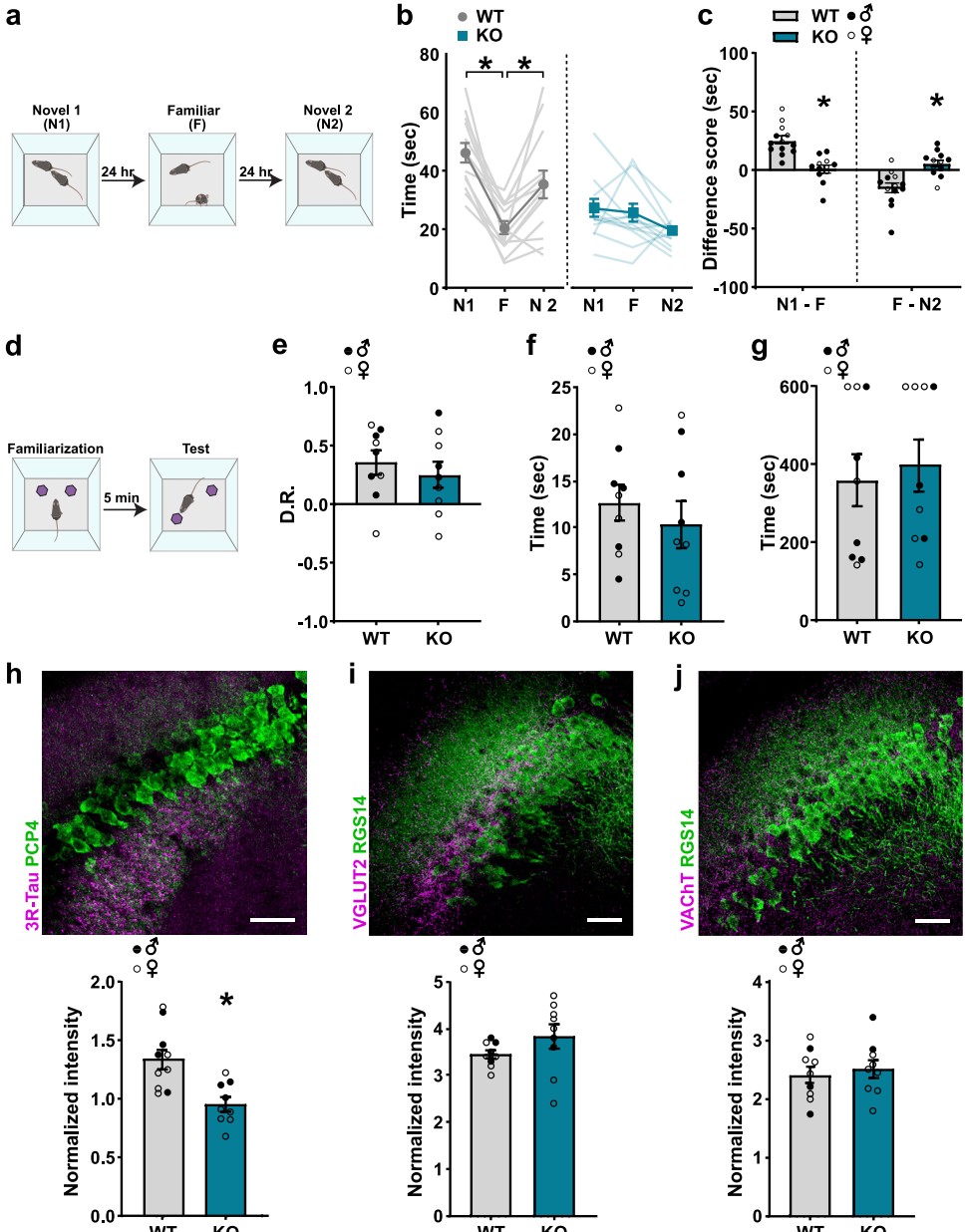

**Fig. 1 | *Shank3B* KO mice have impaired social, but not object location, discrimination and mostly typical CA2 afferents. a** Schematic of social discrimination test. **b** WT mice have lower interaction times for familiar (F) than first novel mice (N1) ($p = 0.0001$) and higher interaction times for second novel mice (N2) than $F$ ($p = 0.0112$). KO mice do not show different interaction times for F compared to N1 or N2 ($n = 13$ for WT and $n = 12$ for KO). **c** Compared to WT, KO mice showed significantly different difference scores (N1 minus F: $p = 0.0001$; F minus N2: $p = 0.000491$; $n = 13$ for WT and $n = 12$ for KO). **d** Schematic of object location test. **e** WT and KO mice showed no difference in discrimination ratios in the object location test, **f** no difference in time interacting with the objects, and **g** no difference in time to reach criterion (E–G, $n = 9$ for each genotype). **h** Top: Confocal image from the CA2 immunolabeled with PCP4 (green) and abGC afferent marker 3R-Tau (magenta). Bottom: Compared to WT, KO mice have lower intensity of 3R-Tau+ afferents in the CA2 ($p = 0.002$) ($n = 10$ for WT, $n = 9$ for KO). **i** Top: Confocal

image of the CA2 immunolabeled with RGS14 (green) and SUM afferent marker VGLUT2 (magenta). Bottom: WT and KO mice have similar intensity of VGLUT2 + afferents in the CA2 ($n = 9$ for each genotype). **j** Top: Confocal image from the CA2 immunolabeled with RGS14 (green) and cholinergic afferent marker VAChT (magenta). Bottom: WT and KO mice have similar intensity of VAChT+ afferents in the CA2 ($n = 9$ for each genotype). Scale bars = 50 μm. See Table S1 for complete statistics. Data presented as mean ± SEM. *$p < 0.05$, two-sided unpaired $t$ tests (**c,e,f,g,h,j**), Mann Whitney $U$ test (**i**), or two-way repeated measures ANOVA with Bonferroni tests (**b**). DR = discrimination ratio; $N$ = novel; $F$ = familiar; KO = *Shank3B* knockout; WT = wildtype; 3R-Tau=3-repeat tau isoform; PCP4 = Purkinje cell protein 4; VGLUT2 = vesicular glutamate transporter 2; VAChT=vesicular acetylcholine transporter; RGS14 = regulator of G protein signaling. Images in **a**, **d** were created using BioRender.com. Source data are provided as a Source Data file.

cell bodies in the dorsal dentate gyrus and found a slight but significantly lower density in the suprapyramidal, but not infrapyramidal, blade of *Shank3B* KO mice (Fig. S5, Table S2).

Next, we considered excitatory projections from extrahippocampal structures to the CA2 region, including glutamatergic inputs from the hypothalamic supramammillary nucleus (SUM) and

cholinergic inputs from the medial septum, both of which have been implicated in either novelty preference or social discrimination[21,23,73]. We examined the intensity of vesicular glutamate transporter 2 (VGLUT2), which is known to label SUM afferents to the CA2[74], and vesicular acetylcholine transporter (VAChT), which is known to label cholinergic afferents[75], and unexpectedly found no differences

between WT and *Shank3B* KO mice in the CA2; the intensity of both markers was similar between groups (Fig. 1i, j, Fig. S4, Table S1).

Within the hippocampus, Shank3 protein is present in excitatory synapses that co-label with the vesicular glutamate transporter 1 (VGLUT1)[74]. VGLUT1 labels afferents from CA3 and entorhinal cortex pyramidal cells[74,76,77] as well as from mature dentate gyrus granule cells[74]. The lateral entorhinal inputs to the CA2 have been causally linked to social discrimination[22] although those from the CA3 and dentate gyrus have not. It remains possible that the developmental disruption of these projections, either alone or in some combination, in this transgenic model may impact the ability of the CA2 to participate in these functions. Since the CA3, entorhinal cortex, and dentate gyrus each directly project to the CA2[78,79], we analyzed intensity of VGLUT1 labeling in the CA2 and found no difference between WT and *Shank3B* KO mice (Figs. S2A, S3A, Table S2). Because Shank3 protein is especially concentrated in mossy fibers[74], we examined intensity of zinc transporter 3 (ZnT3), which labels mossy fibers and their synaptic vesicles[80], in the CA2 and also found no difference between WT and *Shank3B* KO mice (Fig. S2B, Table S2).

Our results suggest that social discrimination dysfunction in *Shank3B* KO mice is not the result of obvious abnormalities in several developmentally generated inputs to this brain region, although an adult-generated afferent population is diminished. The extent to which diminished abGC numbers and afferents arise directly from *Shank3B* KO and contribute to social memory dysfunction remains unknown.

### Increasing CA2 activity improves social discrimination in *Shank3B* KO mice

Using chemogenetics, we tested whether activating excitatory neurons in CA2 would improve social discrimination in *Shank3B* KO mice. Excitatory DREADD virus (AAV-CaMKIIa-hM3D(Gq)-mCherry) or control virus (AAV-CaMKIIa-GFP) was bilaterally injected into the CA2 of WT and *Shank3B* KO mice (Fig. 2a, b). Histological analyses of virus infection revealed robust expression that was largely confined to the CA2 region (Fig. S6) with very little labeling in the adjacent CA1 or CA3 regions. We analyzed mCherry expression in PCP4 + CA2 cells of a subset of mice and found that ~91% of mCherry+ cells were PCP4+, and ~50% of PCP4 + cells were mCherry+. This suggests that the majority of infected cells were pyramidal neurons and about half of the CA2 pyramidal cell population was infected.

Two weeks after CA2 virus infection, mice were tested for social discrimination abilities using a direct social interaction test. Each virus-injected mouse was tested with both vehicle (VEH) and CNO in separate behavior testing. For each behavioral testing bout, VEH or CNO was injected 30 min prior to each novel and familiar mouse exposures. Testing with VEH and CNO were counterbalanced to control for any order effects with a minimum of two days between each test to allow for wash out of CNO (Fig. 2a).

Consistent with our previous results from unoperated mice (Fig. 1), in the VEH trials, WT mice showed typical social discrimination, whereas *Shank3B* KO mice showed impaired social discrimination. WT mice virus groups (control and DREADD virus) injected with VEH showed lower interaction times for familiar mice than novel mice; whereas, KO mice virus groups (control and DREADD virus) injected with VEH showed no such change across trials (Fig. 2c, Table S1). In addition, the difference in time investigating novel and familiar mice was relatively unchanged for *Shank3B* KO mice virus groups and was significantly different than WT mice virus groups (Fig. 2d, Table S1).

As expected, WT mice injected with either control or DREADD virus decreased their interaction times for familiar mice when administered CNO with no difference noted between control and DREADD virus groups (Fig. 2e, Table S1). CNO had no effects on social discrimination in *Shank3B* KO mice injected with control virus, while *Shank3B* KO mice injected with DREADD virus showed significantly greater investigation times with novel mice compared to familiar mice.

The CNO-induced increase in difference between investigation times with novel mice compared to familiar mice was observed in *Shank3B* KO mice regardless of whether they were tested with CNO or VEH first, strongly suggesting that the results are not dependent on order effects (Fig. S7, Table S2). *Shank3B* KO + DREADD virus mice treated with CNO had social discrimination abilities that did not differ from WT mice (Fig. 2e, Table S1). Moreover, following CNO injections, the difference in time investigating novel and familiar mice was not significantly different between the WT + control and DREADD virus groups and the *Shank3B* KO + DREADD virus group, while the *Shank3B* KO mice control virus group was significantly lower (Fig. 2f, Table S1). In addition, the difference score (difference in time investigating novel and familiar mice) changed within subject in response to drug treatment. We found that KO + DREADD mice have higher difference scores following CNO treatment compared to VEH treatment, while all other groups showed no change in difference scores between VEH and CNO treatment (Fig. S8; Table S2).

### Increasing activity in the CA2 to vCA1 pathway improves social discrimination in *Shank3B* KO mice

Silencing the connection between the CA2 and ventral CA1 has been shown to produce social memory deficits in controls[29], including reducing the difference in interaction times with novel and familiar mice. We next explored whether activating CA2 afferents to vCA1 would improve social discrimination in *Shank3B* KO mice. WT and *Shank3B* KO mice received bilateral injections of excitatory DREADD virus (AAV-CaMKIIa-hM3D(Gq)-mCherry) or control virus (AAV-CaM-KIIa-GFP) in the CA2 and were then implanted with bilateral cannula into vCA1 to allow for local and selective excitation of CA2 projecting neurons (Fig. 3a, b). Two weeks after surgery, mice were tested for social discrimination abilities using a direct social interaction test. Each virus-injected mouse was tested with both VEH and CNO in separate behavior testing. For each behavioral testing bout, VEH or CNO was infused into the vCA1 bilaterally 30 min prior to each novel and familiar mouse exposures. Testing with VEH and CNO was counterbalanced to control for any order effects with a minimum of two days between each test to allow for wash out of CNO (Fig. 3a).

Following VEH infusions into vCA1, compared to their interaction times with novel mice, WT + control virus mice showed typical social discrimination as evidenced by their lower interaction for familiar mice, whereas *Shank3B* KO + control virus mice showed no difference in their interaction times for familiar mice (Fig. 3c, Table S1). It should be noted that although the majority of WT + DREADD virus mice treated with VEH in this study showed typical social discrimination, the overall effect was not statistically significant in this group since a few mice showed a reversed social preference (Fig. 3c,d). The reasons for this remain unknown although it should be noted that we and others have occasionally observed control mice that investigate familiar stimulus mice more than novel mice in other studies[22,81–83]. This does not reflect an inability of social recognition, but instead, a preference for social familiarity, rather than social novelty. We included these mice in the analysis although they are biological outliers because they reflect the range of typical mouse behavior. As a result, in this study, while the difference in time investigating novel and familiar was on average lower for *Shank3B* KO than WT mice, the difference between WT and *Shank3B* KO mice did not reach significance using a stringent post hoc comparison that controls for multiple comparisons (Fig. 3d, Table S1).

CNO infusions into the vCA1 had no effect on WT + control virus or WT + DREADD virus mice as demonstrated by their typical social memory (Fig. 3d, Table S1). As expected, *Shank3B* KO + control virus mice infused with CNO into the vCA1 had impaired social discrimination abilities (Fig. 3d, Table S1). However, *Shank3B* KO + DREADD virus mice injected with CNO into the vCA1 displayed typical social discrimination (Fig. 3d). After vCA1 CNO infusion, *Shank3B* KO + control

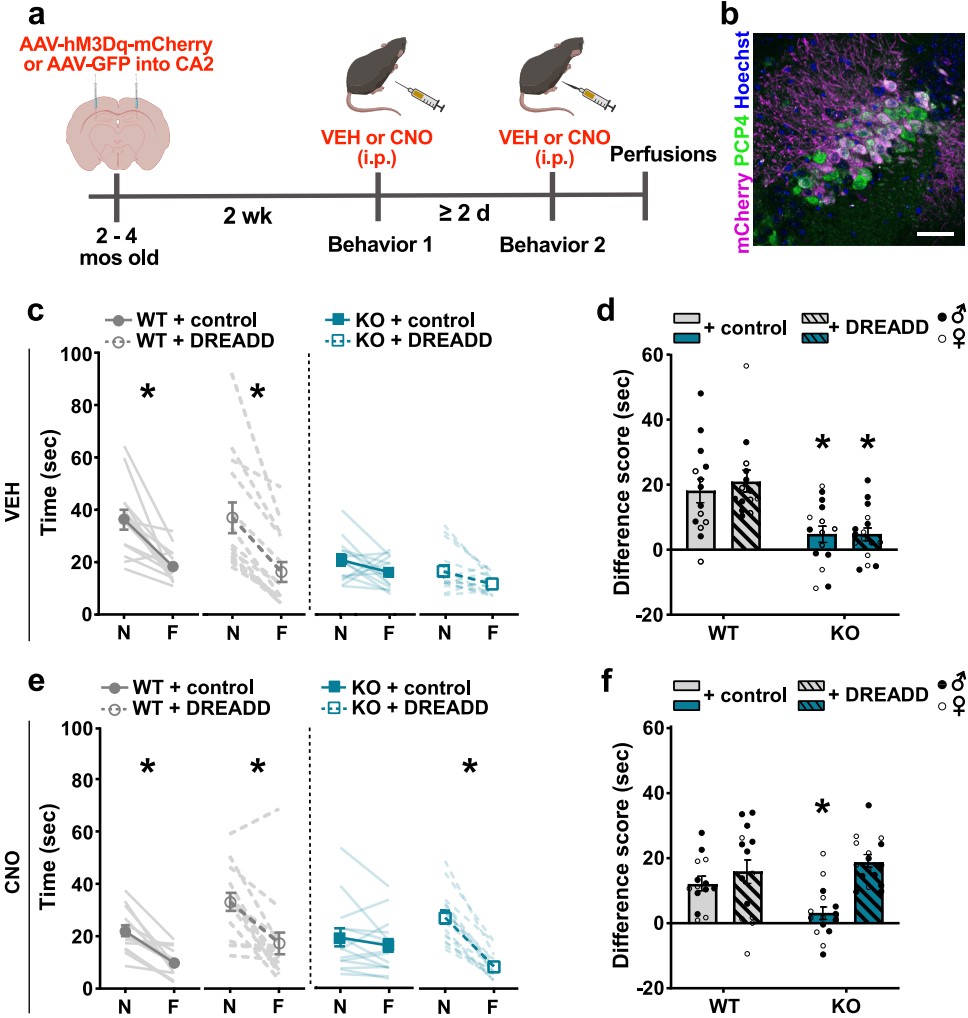

**Fig. 2 | Chemogenetic activation of excitatory neurons in CA2 improved social discrimination in *Shank3B* KO mice. a** Timeline for experiment. **b** Confocal image from the CA2 immunolabeled with PCP4 (green) and mCherry (magenta), showing localized virus infection in CA2 neurons. Scale bar = 50 μm. **c** Following VEH injections, virus-infected WT mice had greater interaction times for novel mice (N) than familiar mice (F) (WT + control virus, *p* = 0.0001; WT + DREADD virus, *p* = 0.0001), while virus-infected KO mice showed no difference in interaction times of N compared to F. **d** Following VEH injections, WT virus groups had higher difference scores (N minus F) than KO virus groups (WT + control virus vs KO + control virus: *p* = 0.0110; WT + DREADD virus vs KO + DREADD virus: *p* = 0.0011). (**c**, **d**, *n* = 14 for WT + control virus, WT + DREADD virus, *n* = 15 for KO + control virus, and *n* = 16 for KO + DREADD virus. **e** Following CNO injections, virus-infected WT mice had greater interaction times for N than F (WT + control virus, *p* = 0.0002; WT + DREADD virus, *p* = 0.0001). Control virus-infected KO mice had no interaction time differences between N and F, while DREADD virus-infected KO mice had greater

interaction times for N than F (*p* = 0.0001). **f** Following CNO injections, WT virus groups had higher difference scores than KO + control virus-infected mice (WT + control virus vs KO + control virus: *p* = 0.0809; WT + DREADD virus vs KO + control virus: *p* = 0.0042) but not KO + DREADD virus-infected mice, WT + DREADD virus or KO + DREADD virus mice. KO + control virus-infected mice had lower difference scores than KO + DREADD virus-injected mice (KO + control virus vs KO + DREADD virus: *p* = 0.0002) (**e**, **f**, *n* = 14 for WT + control virus, WT + DREADD virus, *n* = 15 for KO + control virus, and *n* = 16 for KO + DREADD virus). See Table S1 for complete statistics. Data are presented as mean ± SEM. \**p* < 0.05; three-way repeated measures ANOVA with Bonferroni tests (**c**, **e**); two-way ANOVA with Bonferroni tests (**d**, **f**). N = novel; F = familiar; KO = *Shank3B* knockout; WT = wildtype; VEH = vehicle; CNO = clozapine-N-oxide; DREADD = designer receptors exclusively activated by designer drugs; GFP = green fluorescent protein; PCP4 = purkinje cell protein 4. Images in **a** were created using BioRender.com. Source data are provided as a Source Data file.

virus mice showed no change in time spent investigating novel and familiar mice with difference times that were significantly lower than WT + control virus and *Shank3B* KO + DREADD virus mice (Fig. 3e, Table S1). WT mice virus groups (control and DREADD virus) had lower interaction times for familiar mice than novel mice when infused with CNO, showing no difference between control and DREADD virus groups. *Shank3B* KO + control virus group retained their lack of difference across trials even when infused with CNO, while the *Shank3B* KO mice DREADD virus group infused with CNO showed significantly greater investigation times with novel mice compared to familiar mice. The *Shank3B* KO + DREADD virus group treated with CNO had social discrimination abilities that did not differ from WT virus groups treated with CNO (control and DREADD virus) (Fig. 3e, Table S1). Moreover,

for the CNO infusions, the difference in time investigating novel and familiar mice was not significantly different between WT virus groups and the *Shank3B* KO + DREADD virus group, while the *Shank3B* KO + control virus group was significantly lower (Fig. 3f, Table S1).

## vCA1 theta power does not differ between WT and *Shank3B* KO mice, but increases in *Shank3B* KO mice after chemogenetic activation of the CA2

vCA1 theta (4–12 Hz) power is associated with higher avoidance behavior[84–86] and because compared to WT mice, *Shank3B* KO mice spend considerably less time investigating novel mice, we explored whether *Shank3B* KO mice had higher vCA1 theta power during exposure to novel mice. WT and *Shank3B* KO mice received bilateral

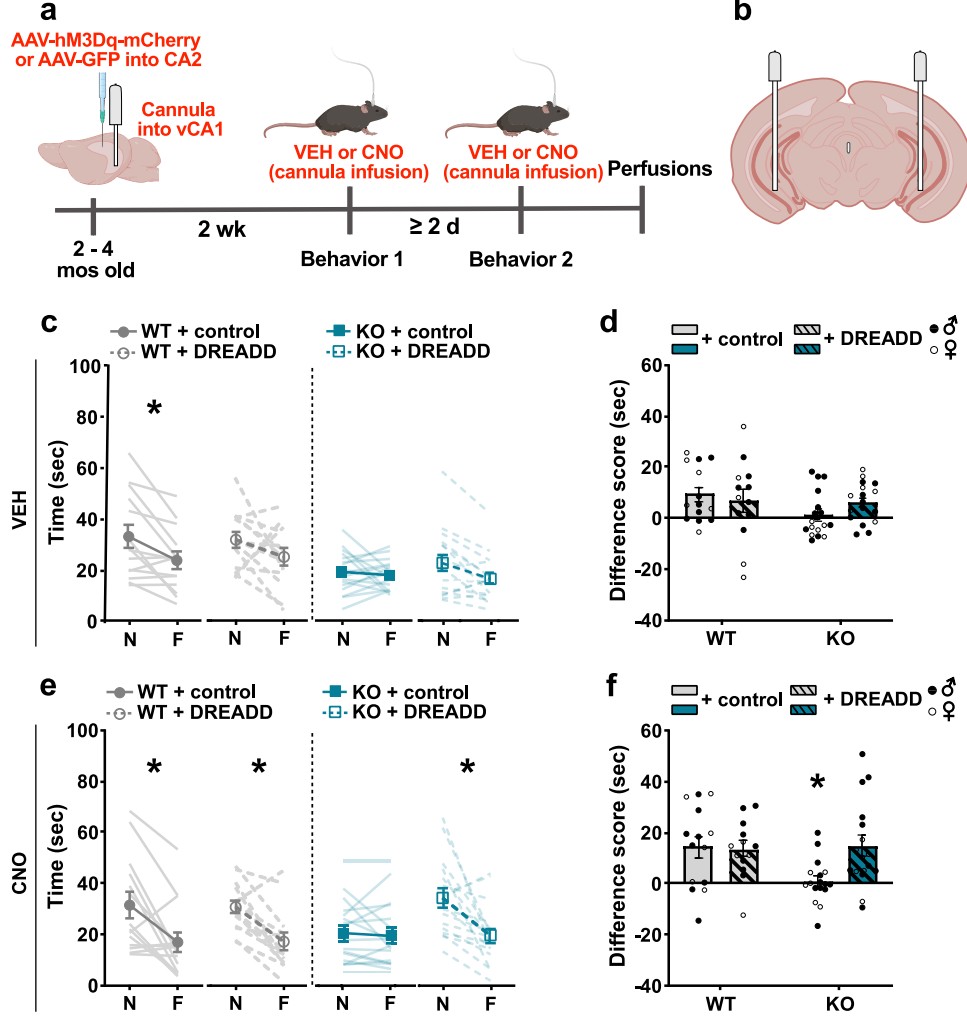

**Fig. 3 | Chemogenetic activation of CA2 excitatory neurons projecting to the vCA1 improved social discrimination in *Shank3B* KO mice. a** Timeline for experiment. **b** Schematic of the cannula placement sites in vCA1. **c** Following VEH infusions into the vCA1, virus-infected KO mice showed no significant difference in interaction times between novel mice (N) and familiar mice (F), while control virus- or DREADD virus-infected WT mice showed a decrease in interaction times between N and F, although the difference in WT + DREADD virus was not statistically significant (WT + control virus, *p* = 0.0270; WT + DREADD virus, *p* = 0.3529). There was no effect of virus group (see Table S1). **d** Following VEH infusions into the vCA1, both WT virus groups had greater difference scores (N minus F) than KO virus groups, although these comparisons were not statistically significant (**c**, **d**, *n* = 14 for WT + control virus, *n* = 13 for WT + DREADD virus, *n* = 17 for KO + control virus, *n* = 17 for KO + DREADD virus). See Table S1 for complete statistics. **e** Following CNO infusions into vCA1, WT virus groups and DREADD virus-injected KO mice had

lower interaction times of F than N (KO + DREADD virus, *p* = 0.0005; WT + control virus, *p* = 0.0024; WT + DREADD virus, *p* = 0.0079), while control virus-injected KO mice did not. **f** Following CNO infusions into vCA1, DREADD virus-injected KO mice had higher difference scores (N minus F) than control virus-injected KO mice (*p* = 0.0302) and no difference from WT mice injected with control virus or DREADD virus mice (**e**, **f**, *n* = 14 for WT + control, *n* = 13 for WT + DREADD, *n* = 17 for KO + control virus, *n* = 17 for KO + DREADD virus). See Table S1 for complete statistics. *\*p* < 0.05; three-way repeated measures ANOVA with Bonferroni comparisons (**c**, **e**); two-way ANOVA with Bonferroni comparisons (**d**, **f**). Data are presented as mean ± SEM. N = novel; F = familiar; KO = *Shank3B* knockout; WT = wildtype; VEH = vehicle; CNO = clozapine-N-oxide; DREADD = designer receptors exclusively activated by designer drugs; GFP = green fluorescent protein. Images in **a**, **b** were created using BioRender.com. Source data are provided as a Source Data file.

injections of excitatory DREADD virus (AAV-CaMKIIa-hM3D(Gq)-mCherry) or control virus (AAV-CaMKIIa-GFP) in the CA2 and were then implanted with recording electrodes into vCA1 (Fig. 4a). We found no difference in vCA1 theta power between WT and *Shank3B* KO mice (Fig. 4c, Table S1), nor did we find a correlation between vCA1 theta power and the time spent investigating novel mice in WT or *Shank3B* KO mice (Pearson's rank correlation coefficient test, WT: *r* = −0.2965, *p* = 0.4384; KO: *r* = 0.1291, *p* = 0.7405). Taken together with our EPM results, these findings suggest that compared to WT mice, low investigation of novel social stimuli observed in *Shank3B* KO mice is not directly related to general avoidance, or to avoidance-associated vCA1 theta power.

Because social stimuli have been shown to increase hippocampal theta power[33], and because chemogenetic activation of CA2

and the CA2-vCA1 pathway increased investigation of novel, but not familiar, mice (Figs. 2e, 3e), we examined whether *Shank3B* KO CA2 DREADD-infected mice treated with CNO showed changes in vCA1 theta power during novel mouse exposure. We found that chemogenetic activation of the CA2 increased vCA1 theta power in the *Shank3B* KO + DREADD virus group, but not in the WT groups (Fig. 4b, c, Table S1). Because dorsal hippocampal theta power has been linked to indices of locomotion[87–90], we examined whether mice with increased vCA1 theta power (i.e., *Shank3B* KO + DREADD virus + CNO) showed increased locomotion. We found a genotype difference in mobility (*Shank3B* KO mice moved less than WT mice), but no difference with CNO treatment in either control virus or DREADD virus mice of either genotype (Fig. S9, Table S2). Taken together, these findings suggest that increases in

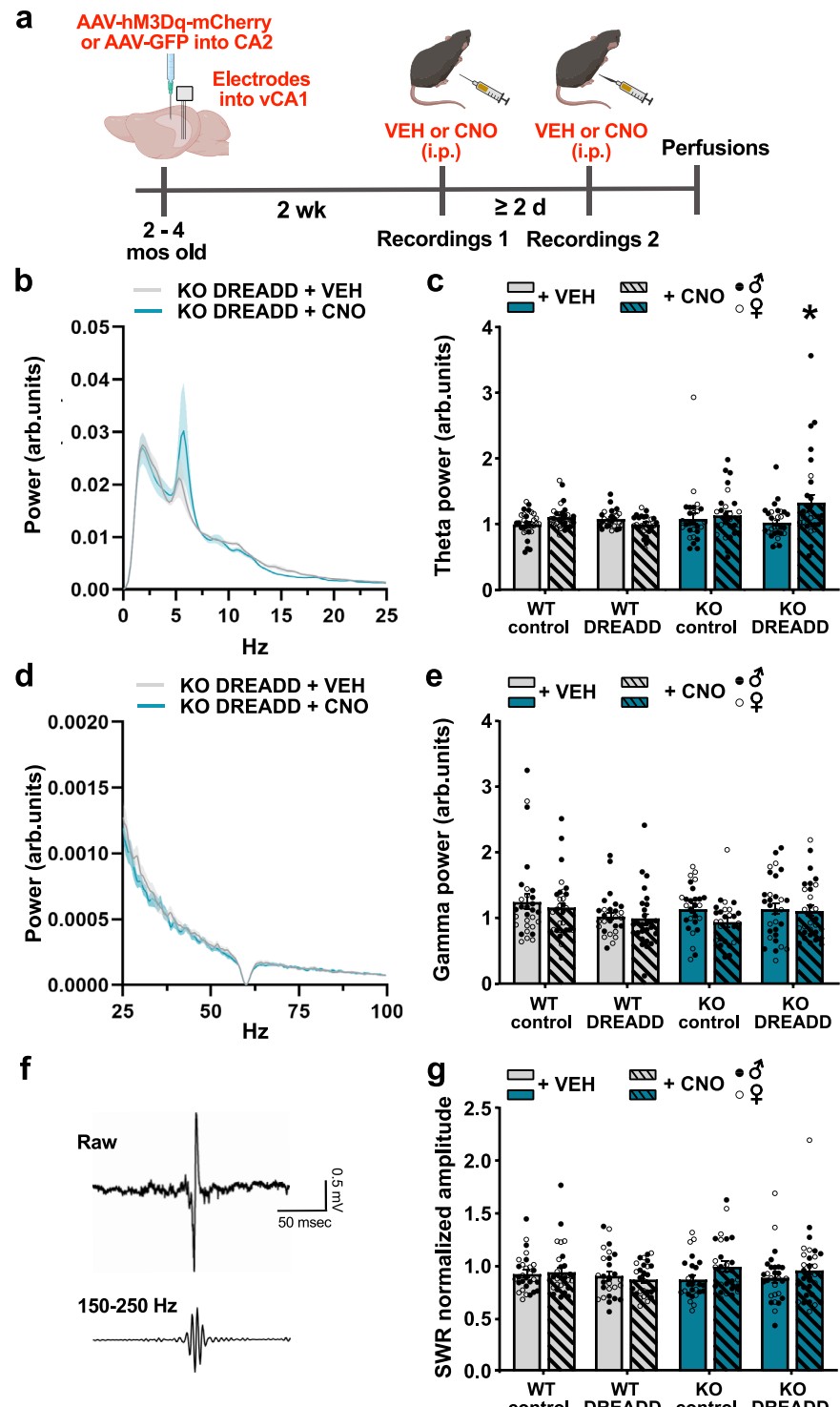

vCA1 theta power in *Shank3B* KO + DREADD virus mice are not related to increased locomotion.

### vCA1 low gamma power does not differ between WT and *Shank3B* KO mice, and remains unchanged after chemogenetic activation of the CA2

Previous studies have shown that low gamma (30–55 Hz) is altered by chemogenetic activation of CA2 neurons in the CA2/proximal dCA1, especially during periods of mobility[91]. Thus, we investigated whether DREADD activation of CA2 neurons altered low gamma power in WT and *Shank3B* KO mice during exposure to novel mice and found no differences across any groups when the data were analyzed overall or

specifically during periods of mobility (Figs. 4d, e, S10, Tables S1, S2). The difference between our findings and the previous report may be due to recording from different areas of the CA1 (dorsal in ref. [19] versus ventral in the present study) or differences in the proportion of CA2 pyramidal cells infected in each study (the percentage of PCP4 + cells that expressed DREADD virus in ref. [19] was greater than 90% versus 51% in the present study).

### SWRs are similar between WT and *Shank3B* KO mice, and not changed with chemogenetic activation of CA2

SWRs have been linked to consolidation and retrieval of spatial memories[25]. The vCA1 has been shown to have increased SWR events in

**Fig. 4 | Chemogenetic activation of the CA2 increased vCA1 theta power in *Shank3B* KO mice, but did not affect gamma power nor SWRs. a** Timeline for experiment. **b** vCA1 power spectra during exposure to a novel mouse (0-25 Hz) of VEH- and CNO-treated KO + DREADD virus-injected mice. Shaded area represents SEM. **c** vCA1 theta power (4–12 Hz) was not different between WT and KO virus groups treated with VEH (see Table S1). KO + DREADD virus mice treated with CNO had higher theta power than KO + DREADD virus-injected mice treated with VEH ($p = 0.0009$), while no other groups had significant differences (n = number of mice with 3 electrodes per mouse; $n = 10$ WT + control virus, KO + DREADD virus; $n = 9$ WT + DREADD virus, KO + control virus). **d** vCA1 power spectra during exposure to a novel mouse (25–100 Hz) of VEH- and CNO-treated KO + DREADD virus-injected mice. Shaded area represents SEM. **e** vCA1 low gamma power (30–55 Hz) was not different between WT and KO virus groups injected with VEH or CNO during novel mouse exposure (see Table S1; $n$ = number of mice with three electrodes per

mouse; $n = 10$ WT + control virus, KO + DREADD virus; $n = 9$ WT + DREADD virus, KO + control virus). **f** Representative example of SWRs (Top: Raw LFP trace. Bottom: Filtered SWR trace) recorded in the vCA1. **g** There were no differences in normalized SWR amplitude after exposure to a familiar mouse between WT and KO control virus- or DREADD virus-infected groups treated with VEH, nor was this changed by CNO-induced activation of the CA2 ($p > 0.05$, see Table S1) ($n$ = number of mice with three electrodes per mouse; $n = 10$ WT + control virus, KO + DREADD virus; $n = 9$ WT + DREADD virus, KO + control virus). See Table S1 for complete statistics. Data are presented as mean ± SEM for error bars and bands. *$p < 0.05$; linear mixed effects ANOVA (**c**, **e**, **g**) with Tukey comparisons (**c**, **g**). Arb.units=arbitrary units; KO = *Shank3B* knockout; WT = wildtype; VEH = vehicle; CNO = clozapine-N-oxide; DREADD = designer receptors exclusively activated by designer drugs; GFP = green fluorescent protein; SWR = sharp wave ripples; Hz=hertz. Images in **a** were created using BioRender.com. Source data are provided as a Source Data file.

the presence of a social stimulus[27] and CA2-generated SWRs, which are known to influence vCA1 SWRs, are necessary for social memory[26]. Given that our behavioral findings suggest diminished function in the CA2-vCA1 pathway in *Shank3B* KO mice, we explored the possibility of alterations in SWRs in the vCA1 that may be rescued by chemogenetic activation of the CA2. We detected SWR events in the vCA1 during baseline or exposure to familiar mice (Fig. 4f). Since SWRs typically only occur during sleep and behavioral immobility[25], we first measured SWR frequency, with SWR number as a function of immobility time. However, *Shank3B* KO mice spend significantly more time immobile than WT mice (WT: 100.8 ± 10.18 s, KO: 184.8 ± 10.29 s, unpaired *t* test, $t_{17} = 5.785$, $p = 0.0001$), and had reduced SWR frequency (WT: 0.6113 ± 0.05623 event frequency, KO: 0.3492 ± 0.01797 event frequency, LME, $t_{17} = 2.770$, $p = 0.0131$), an effect that clearly confounds the SWR frequency analysis. To eliminate this confound, we focused our analysis on SWR number. We first examined SWR measures normalized to baseline during the first minute of familiar conspecific exposure, a time when retrieval of social memories is most likely. We analyzed normalized SWR number, amplitude, duration and time interval between SWRs in this data set. We found no significant differences between *Shank3B* KO and WT mice with VEH or CNO treatment on any of these measures (Figs. 4g, S11, Tables S1, S2). We then examined SWR number, amplitude, duration, and time interval between SWRs during each minute of baseline (3 min) and exposure to a familiar mouse (5 min). Again, we found no difference in SWR number, amplitude, duration, or time interval between SWRs in WT compared to *Shank3B* KO mice during any minute of testing, overall, or following chemogenetic activation of the CA2 (Figs. S12–S15, Table S2). It should be noted that we detected significant differences between WT DREADD VEH vs CNO (SWR numbers at minute 2 of the familiar conspecific trial: CNO was greater than VEH, Table S2), but these effects were not seen during any other time of the trial, nor did they correspond to any of our behavioral findings.

These findings suggest that although WT and *Shank3B* KO mice differ in their ability to discriminate between novel and familiar mice, there were no differences in SWR measures in response to a familiar mouse, nor is there any notable change in response to CA2 activation. Because we applied a standard cut-off for amplitude in our definition of a SWR, it remains possible that some excluded "SWR-like" events were lower in *Shank3B* KOs than WTs. However, this seems unlikely given the similarities in so many SWR measures between genotypes, which suggest that social discrimination differences are not related to our SWR measures.

## Discussion

Our findings suggest that *Shank3B* KO mice have impaired social discrimination, with a major deficit in time spent investigating novel social stimuli, compared to WT littermates. We found no difference between groups in a simple object location task or in avoidance behavior, suggesting that *Shank3B* KO mice have some unimpaired non-social

memory and that differences in novel social investigation seem unlikely due to general avoidance. Because Shank3 is a synaptic scaffolding protein, we next investigated whether multiple developmentally-generated afferents to the CA2 are different in *Shank3B* KO mice compared to WT mice. Unexpectedly, we found that markers of CA2 afferents known to be involved in social discrimination and social novelty detection, including VGLUT2 and VAChT, were not different from WT in *Shank3B* KO mice. In contrast, we found that 3R-Tau, a marker of abGC afferents, was diminished in the CA2 of *Shank3B* KO compared to WT mice. Because the CA2 has been associated with social memory, we next attempted to improve social discrimination by chemogenetically activating excitatory neurons in the CA2 of *Shank3B* KO mice. We found that activating the CA2 with DREADD virus + CNO restored social discrimination in *Shank3B* KO mice to WT levels. We next investigated whether specific activation of projections from the CA2 to the vCA1 would also produce this effect and found restored social discrimination in *Shank3B* KO mice to WT levels. These findings suggest that *Shank3B* KO social discrimination ability can be improved by activating the CA2-vCA1 pathway in adulthood. We next tested whether vCA1 SWRs, which have been linked to memory consolidation and retrieval, differ between WT and *Shank3B* KO mice and found similarities in amplitude, number, duration, and time interval, as well as no change in vCA1 SWRs during chemogenetic manipulation. Although we also detected no differences in vCA1 theta or low gamma power in *Shank3B* KO compared to WT mice for control virus or vehicle treatments, theta power was increased in *Shank3B* KO mice infected with DREADD virus and treated with CNO beyond WT levels. Increased vCA1 theta power was independent of mobility and was observed during exposure to the novel mouse, which is the experience that elicited behavioral change after CA2 activation in *Shank3B* KO mice.

Our behavioral data raise the possibility that low social investigation of novel mice by *Shank3B* KO mice may be the result of greater social avoidance compared to WT mice. However, we observed no evidence of high avoidance behavior of *Shank3B* KO mice on a non-social task, the elevated plus maze, findings that are consistent with some[48], but not all[67], published reports. We also found no evidence of increased vCA1 theta power, an electrophysiological correlate of non-social avoidance behavior[86], in *Shank3B* KO mice displaying low levels of novel social investigation (*Shank3B* KO + control virus, *Shank3B* KO + DREADD virus + VEH), and no correlation between investigation times of novel mice and vCA1 theta power. In fact, the one experimental group that showed elevated vCA1 theta power (*Shank3B* KO + DREADD virus + CNO) had been subjected to a manipulation that increased investigation of novel mice compared to familiar mice. Taken together, these results suggest that low investigation times are not the result of global behavioral inhibition. Low investigation of a novel social stimulus may represent a number of specific deficits, including faulty recognition of a novel stimulus as familiar, inattention to social stimuli, reduced motivation/reward associated with social stimuli, and/or

inability to recognize social novelty, all of which have been reported in humans with ASD[1,51,92–94]. More specifically related to the *Shank3B* mouse model, individuals diagnosed with both Phelan-McDermid syndrome and ASD exhibit reduced social attention and reduced social novelty recognition under certain conditions[51].

Social novelty recognition, along with broader social memory, has been linked to the CA2 region in mice[79]. Electrophysiological studies have shown increased firing of CA2 pyramidal cells in response to a novel social stimulus[95]. Additional studies have identified afferents to the CA2 as being involved in these functions. In particular, projections from the SUM and the cholinergic basal forebrain have been linked to novelty recognition[21,23,83,] whereas those from the vasopressinergic paraventricular nucleus and adult-generated neurons from the dentate gyrus seem more likely associated with memory of familiar social stimuli[71,96]. Despite the fact that Shank3, a synaptic scaffolding protein, is expressed in the hippocampus, we found no differences between WT and *Shank3B* KO mice in markers of several of these afferents to the CA2. These results were unexpected and suggest that compensation for the lack of Shank3 might occur during development, particularly at synapses with high concentrations of this molecule, including those made by mossy fibers, as well as those expressing VGLUT1[74]. In this regard, it may be relevant that Shank3 colocalizes with other synaptic scaffolding proteins (Shank1 and 2)[74], which may enable the development of functional synapses in the absence of Shank3. It is also possible that these pathways are impacted by *Shank3B* KO, but that the level of resolution of our analyses is not sufficient to detect differences. Despite the lack of global abnormalities in CA2 afferents of *Shank3B* KO mice, we did observe a decrease in the afferents from 3R-Tau labeled abGCs in the dentate gyrus. abGCs, like mature granule cells, are known to project to the CA2[70,78] and have been shown to participate in social memory[71]. Although it is possible that the lower numbers of abGCs and their afferents to CA2 contribute to social discrimination deficits in *Shank3B* KO mice, it seems unlikely to be the main contributor, since transgenic reduction of abGCs reduces social memory function without affecting investigation times of novel social stimuli[71]. It remains unknown how *Shank3B* KO affects abGC projections to the CA2 without exerting a measurable influence on the overall mossy fiber projection (stained with ZnT3).

Despite the lack of obvious neuroanatomical differences in afferent projections which may be involved in the recognition of novel social stimuli between WT and *Shank3B* KO mice, it remains possible that synapses between these afferents and the CA2 are affected. Since inputs from the SUM release Substance P, a neuromodulator known to enhance NMDA responses in the CA2[96], it is possible that this connection is atypical in *Shank3B* KO mice due to differences in Substance P release or NMDA receptor trafficking at the synapse. With regard to the latter possibility, however, a study has shown that social deficits in *Shank3B* KO mice are not resolved by treatment with a partial agonist of the NMDA receptor[97]. Future studies investigating the effects of additional NMDA receptor manipulations as well as whether post-synaptic elements of relevant connections are different in *Shank3B* KO mice will be necessary to answer these questions.

Regardless of the lack of global abnormality of several excitatory afferent labels in CA2 of *Shank3B* KO mice, chemogenetic activation of CA2 pyramidal cells in general or the CA2-vCA1 pathway directly in these mice restored social discrimination to resemble that of WT mice. Analysis of social investigation times suggests that the main effect of CA2 excitatory neurons or CA2-vCA1 activation is to increase investigation times of the novel mouse. This effect was only seen in the *Shank3B* KO mice, with no changes observed in the WT mice following identical treatment. These findings suggest that an, as yet, unidentified dysfunction upstream or within the vCA1, potentially of the CA2 and/or CA2-vCA1 pathway, exists in *Shank3B* KO mice, but that stimulating existing circuitry is sufficient to invoke latent function. Thus, although

our experiments did not uncover a mechanism underlying the social deficit in *Shank3B* KO mice, we have identified a manipulation that can restore this behavior to WT levels.

Oscillatory activity in the CA2 and vCA1 regions has been linked to social memories. CA2 SWRs have been causally associated with social memory consolidation[26], raising the possibility that abnormalities in their number, magnitude, or duration might contribute to social memory dysfunction in *Shank3B* KO mice. Because previous studies have shown that SWRs are also involved in retrieval of nonsocial memories[25], we hypothesized that vCA1 SWRs might differ between WT and *Shank3B* KO mice during familiar mouse presentation, a time when social memory retrieval is most likely. However, we found no differences in several SWR measures between WT and *Shank3B* KO mice, as well as no changes in these measures after chemogenetic activation of CA2 excitatory neurons during baseline trials or throughout the familiar mouse exposure trial. It remains possible that vCA1 SWRs might differ between WT and *Shank3B* KO mice but that the parameters of our recordings did not capture this effect. Along these lines, it is relevant to note that a recent study found diminished SWR amplitude in *Shank3B* KO compared to WT mice during the 2 h rest period after social interaction, a time likely important for social memory consolidation[28]. However, since the main difference we noted in social interaction between WT and *Shank3B* KO mice was with investigation of novel social stimuli, which does not involve memory consolidation or retrieval, it seems likely that this social novelty deficit is not dependent on differences in SWRs.

Hippocampal theta and gamma rhythms have been associated with novelty recognition[30,98–103]. Since social stimuli have been shown to increase hippocampal theta and gamma oscillations[30,33,104] and mouse models of social dysfunction have been associated with alterations in hippocampal oscillations at both frequencies[34,35,105], we examined theta and low gamma power in the vCA1 of WT and *Shank3B* KO mice in the presence of a novel mouse. Unexpectedly, we found no differences in either theta or low gamma power in the absence of CA2 activation. Chemogenetic activation of CA2, however, produced an increase in vCA1 theta but not low gamma power in *Shank3B* KO mice. This effect was not observed in WT mice, paralleling the results of our behavioral studies. Taken together, these findings suggest that although theta power does not appear to be atypical in *Shank3B* KO mice in response to novel mice, it is increased under conditions associated with a restoration of novel mouse investigation times to WT levels. Thus, although our studies did not uncover an underlying abnormality of *Shank3B* KO mice in social memory circuitry, elevated vCA1 theta power corresponds with enhanced investigation of novel mice by *Shank3B* KO mice.

The possibility that chemogenetic activation of CA2-induced vCA1 theta power in *Shank3B* KO mice is responsible for increased investigation of novel social stimuli may seem contradictory with reports of causal links between vCA1 theta power and avoidance behavior[86], however, that association has only been observed with nonsocial avoidance, and we found no correlation between theta power and social investigation times in WT or *Shank3B* mice without DREADD virus and CNO. Studies have shown that the vCA1 is heterogeneous anatomically and functionally, with subsets of neurons connected to the hypothalamus/medial prefrontal cortex, amygdala, and nucleus accumbens, each serving different functions, including avoidance, learning, and social behavior respectively[106]. Indeed, theta coherence between hippocampus and different target regions has been reported to differ after exposure to social stimuli versus nonsocial stimuli, with the latter linked to defensive/avoidance behavior[104]. Thus, it seems likely that chemogenetic activation of the CA2 may activate theta oscillations among subsets of vCA1 neurons that have downstream connections with regions associated specifically to social behavior.

CA2 pyramidal cells are known to synapse onto vCA1 pyramidal cells[31] and likely also influence parvalbumin (PV) + interneurons, which

are known to participate in the generation of theta oscillations[107]. Since vCA1 PV + cells have been shown to increase their firing in response to novel social stimuli[108], increased CA2 excitatory input may drive rhythmic firing of PV + interneurons in the theta range, producing activity that is sufficient to enhance investigation of novel mice and facilitate discrimination between them and familiar mice. Whether PV + interneurons in the vCA1 are impacted in *Shank3B* KO mice remains to be determined but if they are diminished in some way, increased excitatory drive from the CA2 might be sufficient to compensate and restore social discrimination abilities. However, since we detected no differences in vCA1 theta power between WT and *Shank3B* KO mice without CA2 activation, any abnormality in vCA1 PV + cell function is likely beyond their ability to generate theta oscillations.

Several studies have shown that *Shank3B* KO mice exhibit lower basal synaptic transmission as well as deficient LTP in the CA1[109-112]. While these effects have only been investigated in the dorsal, not the ventral, CA1, studies in wildtype rodents suggest that vCA1 exhibits LTP, albeit less robustly, than dorsal CA1[113,114]. This raises the possibility that chemogenetically-induced increases in theta oscillations may be sufficient to induce vCA1 plasticity. However, it should be noted that no previous studies have linked vCA1 LTP to social novelty detection or social discrimination.

Collectively, our findings suggest that *Shank3B* KO mice have deficits in social discrimination as a result of reduced investigation of novel social stimuli. Despite the lack of gross morphological abnormality in the CA2 of *Shank3B* KO mice, we found that chemogenetic stimulation of the CA2 and the CA2-vCA1 circuit was sufficient to restore social investigation of novel stimuli. Behavioral restoration in *Shank3B* KO mice with CA2 activation was associated with increased vCA1 theta power, but not low gamma power or alterations in SWRs. These findings suggest that activation of a hippocampal social memory circuit in adulthood is sufficient to restore a behavioral deficit arising from a neurodevelopmental genetic anomaly. The extent to which our results are relevant to humans with ASD remains to be determined.

## Methods
### Animals
All animal procedures were approved by the Princeton University Institutional Animal Care and Use Committee and were in accordance with the National Research Council Guide for the Care and Use of Laboratory Animals. All mice were group housed by genotype and sex in Optimice cages on a reverse 12/12 h light/dark cycle and tested in the dark. Humidity of the room was ~50%. The mice were provided ad lib access to food and water. To generate WT and *Shank3B* KO mice, adult male and female *Shank3B* ± (JAX Stock no. 17688) mice were obtained from The Jackson Laboratory and bred at Princeton University using a heterozygous X heterozygous strategy. At postnatal day 15, mice were genotyped by Transnetyx using real-time PCR. Mixed groups of male and female WT and male and female *Shank3B*-/- null mutant mice were used as test mice. *Shank3B* +/- heterozygous mice that were the same sex and age as the test mice were used as stimulus animals. All mice were group housed by genotype and sex in Optimice cages on a reverse 12/12 h light/dark cycle. For social memory and object location memory behavioral studies, as well as histological studies, 6- to 8-week old mice were used ($n = 9$–13/group). For elevated plus maze studies, 2–5 month old mice were used ($n = 20$/group). For behavioral experiments involving chemogenetic activation of the CA2 and CA2-vCA1 pathway, 2–5 month old mice were used ($n = 13$–18/group). For electrophysiological experiments, 2–5 month old mice were used ($n = 9$–10/group).

### Surgical procedures
Mice were deeply anesthetized with isoflurane (2–3%) and placed in a stereotaxic apparatus (Kopf) under a temperature controlled thermal blanket. The head was levelled using bregma, lambda, and medial-lateral reference points before craniotomy was performed. Each mouse received bilateral injections (15 nl/hemisphere at a rate of 15 nl/min) of either excitatory DREADD virus AAV-CaMKIIa-hM3D(Gq)-mCherry (Addgene viral prep # 50476-AAV5, titer: $1.7 \times 10^{13}$) or control virus AAV-CaMKIIa-EGFP (Addgene viral prep # 50469-AAV5, titer: $4.3 \times 10^{12}$) into the CA2 using the following coordinates from Bregma: −1.82 AP, ±2.15 ML, and −1.67 DV. Both viruses were serotype AAV5. The virus was delivered using a 10 μl syringe with a 33-gauge beveled needle (NanoFil) controlled by a microinjection pump (WPI). The needle remained in place for an additional 5 min after the injection was completed to prevent backflow of the virus upon removal.

For activating the CA2-vCA1 projections, mice injected with DREADD or control virus into the CA2 were implanted bilaterally with a cannula guide extending 4 mm (Plastics One, Cat# C315GS-5/SP) into the vCA1 (−3.5 AP, ±3.45 ML). Dummy cannula (Plastics One, Cat# C315DCS-5/SPC) were inserted into the guides and the guide was lowered to −3.8 mm. Cannula guides were kept in place using meta-bond and dental cement (Bosworth Trim).

For vCA1 recordings, mice injected with control or DREADD virus into the CA2 were implanted unilaterally with a custom-made 3 wire electrode array (Microprobes) into the right hemisphere of the vCA1 (electrode 1, AP: −3.3, ML: 3.45, DV: −3.8 with each electrode separated by 200 nm). Four bone screws were implanted on the skull and one screw was used as a ground. A ground wire was wrapped around the ground screw and covered with metallic paint. Electrode implants were kept in place using Metabond and dental cement (Bosworth Trim). Two to four weeks after surgeries, mice were i.p. injected or cannula infused with either clozapine-N-oxide (CNO) or VEH (see CNO administration) before being tested on behavioral tasks (see Behavioral assays) and/or underwent electrophysiological recordings (see Electrophysiology recordings).

### CNO administration
Each CA2 virus-injected mouse underwent social discrimination testing (with novel and familiar stimulus mice) twice, once after CNO i.p. injection or vCA1 cannula infusion and once after VEH i.p. injection or vCA1 cannula infusion. The order of drug administration (CNO or vehicle) was counterbalanced across groups. Because previous studies have shown that DREADD manipulations of neurons are transient and return to baseline by 10–24 h post-CNO injections[91,115,116], mice were given a minimum 2-day rest period between CNO and VEH tests. 30 min prior to both the novel (trial 1) and familiar (trial 2) stimulus mouse exposure, test mice received CNO or VEH i.p. injections or vCA1 cannula infusions. For systemic administration of CNO, CA2 virus injected mice received i.p. injections of 1 mg/kg of CNO (dissolved in DMSO and then suspended in saline) or VEH. To activate the CA2-vCA1 pathway, CA2 virus injected mice with implanted cannula received cannula infusions of CNO into the vCA1 under light isoflurane anesthesia (2%). After the dummy cannula was removed, an internal cannula projecting 0.8 mm (Plastics One, Cat# C315IS-5/SPC) from the tip of the guide cannula was inserted. 1 μl of CNO (2 μg/μl of CNO dissolved in DMSO and then suspended in saline)[117] or VEH was infused per hemisphere over 1 min into the vCA1 using a syringe pump (Harvard apparatus) mounted with a 1 μl syringe (Hamilton). The internal cannula remained in place for 1 additional minute after the infusion was completed to allow for diffusion of the drug. Mice were returned to their home cages and resumed typical ambulatory activity from the light anesthesia within 5 min.

### Behavioral assays
All behavioral testing occurred during the active cycle for mice (dark cycle). The testing arena was an open field box or an elevated plus maze (see below for details). All behavior was analyzed manually from videotapes by researchers blind to the treatment condition. Since WT and KO mice look virtually identical to one another and all mice in a

given experiment had similar manipulations (e.g., bilateral CA2 infection with or without bilateral vCA1 cannula or unilateral vCA1 electrode), a numerical ear tag code, which was not decoded until the behavioral analyses were complete, ensured unbiased scoring. Video recordings were made using a digital HD video camera recorder (Sony Hanycam HDR-XR150) with standard definition high-quality settings (30 frames per second).

**Social discrimination memory testing.** To assess social discrimination, two versions of the direct social interaction test were adapted from previously established protocols[17,71,82,118]. Each version of this test consisted of two or three social stimulus trials, each separated by 24 h. For behavioral characterization studies, each mouse was tested once with a three social stimulus trials paradigm. For virus manipulation studies, each mouse underwent social discrimination testing two times, each time with either VEH or CNO treatment (order of injections or infusions was counterbalanced across groups) with VEH and CNO treatment separated by at least a 2-day wash-out period. All habituation and testing was done under dim light (10–15 lux). Prior to the test beginning, mice were acclimated to the behavior testing room for at least 30 min and then also habituated to the testing box for 5 min prior to the first social stimulus trial. The testing was conducted in low lighting in an open-field box (23 × 25 × 25 cm). For the three-social stimulus trial paradigm, the test mouse and a never-before-encountered mouse (Novel 1, trial 1) were placed together in the testing box and allowed to interact for 5 min. After this interaction period, the test mouse was returned to their home cage for 24 h and then placed back into the testing box the previously encountered mouse (Familiar, trial 2). 24 hours after the second trial, the test mouse was introduced to a new, novel mouse (Novel 2, trial 3). For experiments involving virus manipulations, mice underwent two social stimulus trials during which the test mouse was introduced to a novel mouse (Novel 1) in trial 1, followed by the same novel mouse (Familiar) in trial 2. Sex-matched non-littermate heterozygous mice were used as stimulus mice for social discrimination testing. For each trial, the interaction time of the test mouse with the stimulus mouse was measured from video recordings. Social investigation was defined as the test mouse directing its snout toward the stimulus mouse's anogenital region or body < 1 cm away, following, or allogrooming that was initiated by the test mouse.

**Object location memory.** To assess a form of non-social memory, the object location test was used[66]. The testing was conducted in low lighting (10–15 lux) in an open-field box (23 × 25 × 25 cm). For 5 min, twice per day, for 3 days, mice were habituated to the testing arena as previously described[82,119,120]. Mice were habituated to the objects on the third day of habituations. Two objects, each <8 cm in height or width, with varying surfaces for the mice to explore, such as LEGO toys and plastic clips, were used. Different, but similar size and shape, objects were used for habituation and testing. In the familiarization trial, two identical objects were positioned on the same side of the testing box (6 cm away from the walls and 10 cm between each other). Mice were free to explore the objects until they reached 30 s of cumulative exploration time with both objects or up to a maximum of 10 min elapsed. The criterion for object exploration was directing their nose at 2 cm or less distance from the object. After the familiarization trial, mice were placed in their home cage for 5 min. Between trials, one object (moved object) was rotated 180° and moved to the opposite wall of the chamber so that it was diagonal to the first object, while the other object was not moved (familiar object). The moved object was counterbalanced throughout testing. For the test trial, mice were returned to the testing arena and were free to explore the objects for 2 min. The object exploration times were scored manually from video recordings. A discrimination ratio (DR) was calculated for each mouse as follows: (Time exploring moved object – time exploring familiar object)/total time exploring objects).

**Elevated plus maze.** Mice were placed on an elevated plus maze that consisted of an elevated (50 cm) plus-shaped track with two arms that were enclosed with high walls (30 cm) and two open arms that had no walls and illuminated to 200 lux. All arms were 50 cm in length. During testing, the mouse was placed in the center of the maze and allowed to explore for 5 min. The number of entries into the open and closed arms and time spent in the open arms, closed arms, and center was measured for each mouse from video recordings.

## Electrophysiology recordings
Local field potentials (LFPs) were recorded using a wireless head stage (TBSI, Harvard Biosciences). Mice were habituated to the weight of recording headstage using a custom headstage with equivalent weight in the home cage for 5 min the day before the first test and in the testing box for 5 min on day of testing. The two social stimulus trial paradigm was used. For each trial, 30 min after drug administration, the test mouse was placed with the stimulus mouse and LFPs recorded continuously for 5 min. To get a baseline measurement, LFPs were also recorded for 3 min in the testing box prior to exposure to the social stimulus each day. The data were transmitted to a wireless receiver (Triangle Biosystems) and recorded using NeuroWare software version 3.0 (Triangle Biosystems).

## Electrophysiological analyses
All recordings were preprocessed using Neuroexplorer software version 5.21 (Nex Technologies) and analyzed using an in-house script (Python). For theta and gamma analyses, continuous LFP data were notched at 60 Hz and band-pass filtered from 0 to 100 Hz. To normalize theta and gamma oscillations, the sum of power spectra values from 0 to 100 Hz were set to equal 1. To obtain power estimates within theta (4–12 Hz) and low gamma (30–55 Hz) bands, the summed power across time for the entire session within each frequency was taken. For SWR analyses, continuous LFP data were notched at 60 Hz and band-pass filtered from 150 and 250 Hz. Signals were then Hilbert transformed and z-scored. SWR events were detected using a custom python script[121]. SWR events were defined in the analysis as instances where the signal exceeded three standard deviations across a rolling-average amplitude threshold for at least 15 ms. The total number of events for each recording was then quantified and exported to an excel sheet for statistical analysis. To determine SWR event frequency, the number of SWRs detected were normalized to immobility time (defined as quiet wakefulness where mice do not move except to groom). The mean amplitude and duration of the detected SWRs were calculated across the recording session. Theta and gamma power were analyzed across the 5 min of trial 1, during exposure to the novel conspecific, and normalized by dividing by the baseline trial. SWR numbers, amplitude, and duration were analyzed across the first 1 min of trial 2, during exposure to the familiar conspecific, and normalized by dividing by the baseline trial. Minute-by-minute analyses of SWRs were also performed during the 3 min of baseline and 5 min of exposure to the familiar conspecific for SWR number, amplitude, duration, and interval of time between SWRs.

## Histology
Mice were deeply anesthetized with Euthasol (Virbac) and were transcardially perfused with cold 4% paraformaldehyde (PFA). Extracted brains were post-fixed for 48 h in 4% PFA at 4 °C followed by an additional 48 h in 30% sucrose at 4 °C for cryoprotection before being frozen in cryostat embedding medium at −80 °C. Hippocampal coronal sections (40 μm) were collected using a cryostat (Leica). Sections were blocked for 1½ h at room temperature in a PBS solution that contained 0.3% Triton X-100 and 3% normal donkey serum. Sections were then incubated overnight while shaking at 4 °C in the blocking solution that contained combinations of the following primary antibodies: mouse anti-three microtubule-binding domain tau protein

(3R-Tau, 1:500, Millipore, Cat# 05-803), rabbit anti-Purkinje cell protein 4 (PCP4, 1:500, Sigma-Aldrich, Cat# HPA005792), rat anti-mCherry (1:1000, Invitrogen, Cat# M11217), mouse anti-regulator of G protein signaling 14 (RGS14, 1:500, UC Davis/NIH NeuroMab, Cat# 75−170), rabbit anti-zinc transporter 3 (ZnT3, 1:500, Alomone labs, Cat# AZT-013), rabbit anti-vesicular glutamate transporter 2 (VGLUT2, 1:500, Synaptic Systems, Cat# 135 403), rabbit anti-vesicular glutamate transporter 1 (VGLUT1, 1:250, Invitrogen, Cat# 48-2400), rabbit anti-vesicular acetylcholine transporter (VAChT, 1:500, Synaptic Systems, Cat# 139 103). For 3R-Tau immunohistochemistry, sections were subjected to an antigen retrieval protocol that involved incubation in sodium citrate and citric acid buffer for 30 min at 80 °C prior to blocking solution incubation. For all primary antibody reactions, washed sections were then incubated for 1½ h at room temperature in secondary antibody solutions that contained combinations of the following secondaries: donkey anti-rat Alexa Fluor 568 (1:500, Abcam), donkey anti-mouse Alexa Fluor 568 or 647 (1:500, Invitrogen), or donkey anti-rabbit Alexa Fluor 488 (1:500, Invitrogen). Washed sections were then counterstained with Hoechst 33342 for 10 min (1:5,000 in PBS, Molecular Probes), mounted onto slides, and coverslipped with Vectashield (Vector labs). Slides were coded until completion of the data analysis. Sections through the ventral hippocampus from cannula and electrophysiology studies were stained for Hoechst 33342 in order to verify accurate cannula and electrode placement.

### Histological analysis

**Verification of CA2 infection, vCA1 cannula, and vCA1 electrode placement.** Only mice with evidence of CA2 infection (control or DREADD virus) were included in behavioral and electrophysiological analyses. Virus infection in CA2 was largely confined to this region with minimal expression noted in neighboring CA1 or CA3 (Fig. S6). All mice in the systemic CNO study (Fig. 2) had CA2 virus expression. In the cannula study (Fig. 3), one control virus KO female was excluded from the analysis and in the electrophysiology study (Fig. 4), one DREADD virus WT female was excluded from the analysis because there was no evidence of CA2 infection. In the cannula study, all mice with CA2 virus infection had histological evidence of cannula placement in the vCA1. In the electrophysiology study, all mice showed histological evidence of electrode implantation in the vCA1.

**Optical intensity measurements.** Z-stack images of the CA2 and corpus callosum were taken on a Leica SP8 confocal using LAS X software version 35.6 and a 40x oil objective. The CA2 was defined by PCP4 or RGS14 labeling. Collected z-stack images were analyzed for optical intensity in Image J (version 2.9.0). A background subtraction using a rolling ball radius (50 pixels) was applied to the image stacks. A region of interest (ROI) was drawn and the mean gray value was collected throughout the image stack. In the CA2, the ROI was confined to the stratum lucidum for 3R-Tau and ZnT3, the stratum radiatum and lacunosum moleculare for VGLUT1, and the pyramidal layer and stratum oriens for VGLUT2 and VAChT. The mean gray value of the ROI was calculated for each z-slice and the maximum mean gray value for each z-stack was taken. That maximum of the CA2 ROI was divided by the maximum of the corpus callosum ROI for each section. Each brain's normalized intensity was the average of 3 sections.

**Cell density and percentage measurements.** The number of 3R-Tau+ cells were counted in the dorsal dentate gyrus of the hippocampus on 4 neuroanatomically matched sections using an Olympus BX-60 microscope with a 100× oil objective. The counts for the suprapyramidal blade and infrapyramidal blade of the dentate gyrus were analyzed separately and area measurements were collected using Stereo Investigator software (version 11.03). The density of 3R-Tau was determined by dividing the total number of positively labeled cells by

the volume of the subregion (ROI area multiplied by 40 μm section thickness). To determine the extent and cell types of CA2 DREADD infection, we counted the numbers of mCherry+ cells, PCP4+ cells, and mCherry+/PCP4+ cells in a subset of DREADD virus infected mice ($n = 5$, 4 sections per mouse). The percentage of PCP4+ cells that express mCherry were determined, as well the percentage of mCherry+ cells that express PCP4.

### Statistical analyses and reproducibility

For histological analyses, data sets were analyzed using an unpaired two-tailed Student's $t$ test or Mann Whitney $U$ tests. For behavioral analyses involving two group comparisons, data sets were analyzed using either unpaired two-tailed Student's $t$ tests or a repeated measures two-way ANOVA, as appropriate. For behavioral analyses involving virus manipulations, data sets were analyzed using either a two-way ANOVA or a repeated measures three-way ANOVA, as appropriate. Bonferroni post hoc comparisons were used to follow up any significant main effects or interactions of the ANOVAs. Pearson's correlation coefficient test was used to analyze the association between theta power and social investigation times. Because electrophysiological measurements were taken from multiple electrodes within each mouse, these data were analyzed with linear mixed-effects ANOVAs using the lme4 R package[122]. The level of the measurement was explained by drug, virus, genotype, the three two-way interactions, the three-way interaction, and a random effect of mouse. Tukey post hoc comparisons were used to follow up any significant main effects or interactions using the emmeans R package[123]. All data sets are expressed as the mean ± SEM on the graphs and statistical significance was set at $p < 0.05$. GraphPad Prism 9.2.0 (GraphPad Software), Excel 16.38 (Microsoft), or R studio were used for statistical analyses. All graphs were prepared using GraphPad Prism 9.2.0 (GraphPad Software). Statistical values ($n$ sizes, $p$ values, and statistical test) are reported in the figure legends or supplementary tables.

There were no planned attempts to reproduce findings of the paper, but there were internal replications of *Shank3B* KO social discrimination deficits in three experiments (Figs. 1, 2, 3), and in reversal of this deficit with DREADD virus and CNO (systemic and cannula infused, Figs. 2,3). Staining patterns for each afferent marker were similar across all mice included in the study (Figs. S3a, b, 4a–c), and viral infection/expression of reporter genes (Figs. 2b, S6) was similar in all mice included in experiments shown in Figs. 2–4.

### Reporting summary

Further information on research design is available in the Nature Portfolio Reporting Summary linked to this article.

## Data availability

The raw electrophysiology data generated in this study are available in the figshare database: https://figshare.com/projects/Cope_et_al_2023_vCA1_LFP_data_Shank3B/158552. The raw histology data generated in this study is available in the figshare database: https://figshare.com/projects/Cope_et_al_2023_CA2_Shank3B_ZnT3/159908, https://figshare.com/projects/Cope_et_al_2023_CA2_Shank3B_vGLUT2/159905, https://figshare.com/projects/Cope_et_al_2023_CA2_Shank3B_-_vGLUT1/159851, https://figshare.com/projects/Cope_et_al_2023_CA2_Shank3B_VAChT/159824, https://figshare.com/projects/Cope_et_al_2023_CA2_Shank3B_-_3R-tau/159818. Behavioral data are available from the corresponding author upon request. Source data are provided with this paper.

## Code availability

Custom python script is available at https://github.com/einaraz/SharpWaveRipple. Laham BJ, Zahn E (2023) einaraz/SharpWaveRipple: V0.0.2 (v0.0.2). Zenodo https://doi.org/10.5281/zenodo.7592606.

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

## Acknowledgements

This work was supported by the National Institutes of Health, NIMH 1R01 MH118631-01 to E.G. and NSF GRFP 2021318039 to R.C.W. The authors thank Monica Hanani and Kristen A. Pagliai for assistance with histological analyses, Marieke Jones for her assistance with the LME analyses, Einara Zahn for assistance with the Python script, and Biorender for assistance with the figure schematics.

## Author contributions

E.C.C. designed the experiments, carried out the experiments, analyzed and interpreted the data, and wrote the paper. S.H.W. carried out the experiments, analyzed the data, and edited the paper. R.C.W. carried out the experiments and edited the paper. I.R.G. carried out the experiments, analyzed the data, and edited the paper. B.V. carried out the experiments and edited the paper. B.J.L. created the code and edited the paper. E.G. designed the experiments, analyzed and interpreted the data, and wrote the paper.

## Competing interests

The authors declare no competing interests.
