## [Peer Review File · Nature Communications]

Activation of the CA2-ventral CA1 pathway reverses social discrimination dysfunction in Shank3B knockout miceREVIEWER COMMENTS

Reviewer #1 (Remarks to the Author):

This impressive manuscript investigates hippocampal neural circuitry underlying social discrimination deficits in Shank3B knockout mice. The authors conducted elegant chemogenetics studies which revealed that activating excitatory neurons in the CA2 region, and activating CA2 afferents to vCA1, restored social discrimination two weeks later in Shank3B knockout mice. The social recognition improvement was accompanied by electrophysiological increases in vCA1 theta power in the Shank3B knockouts. Another intriguing finding was a lower intensity of 3R-Tau+ mossy fibers in the CA2 of Shank3B mutant mice.

Methods for the three trial direct social interaction test for recognition memory, and for DREADD administration, are generally well described. Appropriate statistical analyses were used for the behavioral data. Sex differences were not detected. A control comparison using object location memory confirmed that non-social recognition memory was intact in both genotypes. Elevated plus-maze experiments confirmed similar anxiety-related behaviors in WT and Shank3B knockouts, ruling out general avoidance as a potential artifact.

The authors are to be complimented on their thorough experimental design, and their insightful interpretations of the many interesting results.

Minor additions are needed to further improve the manuscript:

1. In the Methods section, first paragraph describing Animals, please state explicitly that the subject mice were null mutant knockouts (Shank3B -/-), or if they were heterozygotes (Shank3B +/-).
2. Please provide considerably more details about the methods used to score the “direct sniffing of the stimulus mouse’s anogenital region or body, following, or allogrooming that was initiated by the test mouse.” (page 18). Include information about exactly how the rater was kept blind to genotype and treatment condition.
3. Please describe the object(s) used for the object location memory test.
4. As described in paragraph 4 of the Introduction, many aspects of social behavior are impaired in people with autism spectrum disorder. However, usually there is not a problem in remembering specific faces if the subject has been able to look directly at the new person’s face sufficiently. Premising a manuscript about social discrimination on autism symptoms is an overreach.

The first two sentences state: “Social memory is an important ability that gives rise to adaptive social interactions. Social memory dysfunction is a characteristic of several neuropsychiatric disorders,

including autism spectrum disorder (ASD), schizophrenia, and major depressive disorder (Williams et al., 2005; Porcelli et al., 2019; Schafer and Schiller, 2019).” This statement will require considerably more descriptions and relevant references to convince the reader that there is evidence for a serious social recognition memory dysfunction in people with autism.

Reviewer #2 (Remarks to the Author):

The study by Cope et al tests the hypothesis that CA2 neuron dysfunction is (at least in part) responsible for social discrimination phenotypes seen in Shank3b KO mice. This work builds upon several previous studies in the last 5-10 years demonstrating that acute or chronic silencing of CA2 neurons in male mice disrupts social novelty discrimination. The dCA2-vCA1 is a well described obligatory circuit for the expression of social recognition memory (PMID: 30301899 PMID: 35811321). There are several extrahippocampal CA2 afferents that have been shown to influence social recognition memory (SuM, PVN, MSDB, LEC), but not all are necessary for the expression social memory. There have been a few studies identifying CA2 specific neuron impairments as a substrate for social memory dysfunction in mouse models (22q11DS, NLG3 KO, sorCS2 KO), but the field is relatively new, and the current body of work adds to this growing and exciting field. There are also several papers investigating social deficits and relevant circuits in Shank3b KO mice (PMID: 32024781, PMC5472997, PMID: 31061091) although none, to my knowledge, that investigate area CA2. Thus, the topic addressed in Cope et al is novel, timely and of broad interest.

Using primarily chemogenetic activation approaches the manuscript describes the main finding that activation of CA2 neurons or terminal-specific activation of the dCA2-vCA1 circuit restores social approach and social discrimination behaviors in Shank3b KO mouse. The authors provide histological and local field potential data to exclude the possibility that social behavior deficits in Shank2b mice are due to disruption in (a subset of) anatomical inputs to CA2 or to differences in high frequency oscillations (low gamma and SWR), respectively. However, the authors do not uncover an underlying mechanism (aside from correlative theta activity) supporting how CA2 neuron activity causally promotes social behavior in Shank2b mice, which limits the excitement over the otherwise well designed and executed study. It is not clear whether there is a CA2 neuron deficit in Shank3b mice that contributes to the social phenotype, but the data clearly show that CA2 activation can promote or improve social behavioral phenotypes in these mice. Differentiating between these two points is an important distinction that warrants more discussion.

Specific comments

The experiments appear to be done with technical rigor and in a sufficient number of male and female mice to support the main funding of the paper. The use of both sexes is welcomed as there is a paucity of social behavioral data after CA2 manipulations in female mice.

Fig 1 B it is not clear what the asterisk next the wt is indicating—the overall effect of genotype x trial? In

general, the positioning of asterisks for significance was a bit confusing throughout the figures.

Fig 1 H-J: red/green merged images are not color blind friendly. Please edit the color scheme and/or show the individual channels separately so the reader can see the quantified staining in question. Representative images per genotype are also welcomed, especially for J (3R-Tau) where there is a significant decrease in MF staining.

I encourage the authors to avoid direct comparisons of sociability and interaction times between WT and KO mice, such as on page 5 results mid paragraph, “they [KO] also had lower novel interactions times as compared to WT, perhaps indicating reduced sociability”.

The authors who established the initial version of the task that is slightly varied here (3-chamber sociability and social novelty task) states in PMC4904775 that “Sociability is a yes-or-no phenotype, obtained by comparing two sides within each group. Sociability is NOT a graded parameter for quantitatively comparing chamber times across groups.” This is underscored by later results (Sup Fig 4) demonstrating that overall mobility/locomotion/ exploration is different between KO and WT, making the comparison difficult to interpret.

If the authors wish to report cross group comparisons on direct measures (i.e., exploration time), please report the F and p stats for the main interaction effect of genotype x trial within the text to justify it (provided in sup table 1). Alternatively, basing conclusions on the statistical result described in Fig 1C is more appropriate, while also acknowledging the limitations of the hypomobility phenotype in Shank3b KO mice as previously shown (PMC5472997).

While the rationale for disrupted afferents to CA2 as a potential locus for CA2 dysfunction resulting in social deficits is logical, the intrahippocampal inputs listed (EC/DG/CA3) are not directly linked to social memory. EC to DG involvement in social memory (Leung et al 2018) has been reinterpreted based on the findings reported in Lopez-Rojas et al and Dang et al (<https://doi.org/10.1007/s12264-021-00813-6>) that suggest the impairment in initially attributed to EC-DG was due to unintended inhibition of EC-CA2 fibers of passage. Chiang et al implicated ventral CA3 to social memory, and to my knowledge ventral CA3 does not project to dorsal CA2. Lopez-Rojas et al also showed that DG circuits (DG-CA3-CA1, DG-CA2-CA1, DG-CA3-CA2-CA1) are dispensable for social memory. I recommend refocusing the rationale without linking the afferents mentioned directly to social memory and highlighting the (lack of) differences in social memory relevant inputs from SuM and MSDB.

While I appreciate the fact that if gross anatomical differences in inputs were uncovered it would likely have functional implications, however because few were found, it begs the question as to whether the inputs present in normal numbers are functionally impaired, perhaps in NMDAR trafficking at the synapse. This limitation should be addressed.

Fig 2B what percentage of PCP4+ cells were DREADD+? Based on the image, it doesn't appear to be 100% and could explain differences across CA2 Gq DREADD studies (i.e., Alexander et al 2018)

Fig 3C No mention of the fact that WT DREADD vehicle controls failed to discriminate novel vs familiar.

Fig 4 were theta/gamma epochs measured during periods of mobility or quiet rest/investigation of social stimulus? There should be larger effect on gamma during periods of mobility. I believe the authors intend to cite Alexander et al 2018 not 2016 as used throughout. Alexander et al recorded from intermediate CA1 with CA2 Gq DREADDs. Can the authors present representative spectrograms with mobility data in addition to the power measurements? Interestingly, in a previous EEG study, an increase in gamma power was detected in Shank3b mice (PMC5472997), although there could be differences based on region.

Reviewer #3 (Remarks to the Author):

The authors have successfully treated social memory abnormalities in Shank3-KO mice, a mouse model of autism, by manipulating the activity of specific memory circuits in the hippocampus, which is interesting and highly promising research. Improvements are listed below.

Major Points

1. The authors claim to manipulate CA2 neuronal activity by microinjecting AAV specifically into the CA2 region (Figure 2B), but CA2 is a very small region, so it is extremely difficult to control without using CA2-Cre mice. The extent to which DREDD expression was observed in CA3/CA1 and the extent to which DREDD was not expressed in CA2 neurons should be examined. Presumably, the authors have confirmed the expression of AAV in the CA2 region in all mice that have completed behavioral testing, but can the data show this?
2. Smith et al., (Mol. Psy. 21:1137, 2016) show that social memory lasts a week when CA2 neurons are hyperactivated. Therefore, in the case of the experiment in Figure 2, there should be more than one week between Behavior 1 and Behavior 2. If not, it is necessary to distinguish between the group that received CNO at Behavior 1 and Behavior 2, and carefully discuss it.
3. The authors make a very generalized claim saying that "SWRs are similar between WT and Shank3B KO mice" (l. 325), even though only the first minute of Trial 2 was included in the analysis, which appears to be arbitrary criteria. More comprehensive data (e.g., SWR numbers, amplitude, duration, frequency [Hz], travel distance, immobility time minute by minute during the social interaction, as well as the baseline trial) would be helpful in understanding if presented. Additionally, in the Discussion, the authors carefully mention the limitations of their analysis, while the abstract uses the very vague sentence; "we observed no differences in these measures between WT and Shank3B KO mice in the vCA1 in response to social stimuli". This section is extremely misleading and should be improved.
4. SWR frequency (l. 337) should be expressed in more precise units, such as the number of SWRs per

second during the immobile time. I am wondering why the authors excluded this most striking phenotype (~50% decrease of SWR occurrence in KO mice!) from their main claims. Is it conceivable that the decrease of SWR is attributable to the diminished SWR amplitude as a result of the exclusion of the SWRs below the threshold of 3SD?

5. It is not clear how “SWR numbers, amplitude, and duration were ... normalized by dividing by the baseline trial” (ll. 637-639). Does this mean that SWRs during Trial 2 were almost three times larger than those recorded during the baseline trial in amplitude (as shown in Figure 4G; which would hardly be the case)?

6. The authors implanted a custom-made 3 wire electrode array and analyzed each electrode as separate samples, which should be statistically treated as a single sample. Also, it should be stated how the authors confirmed that the tips of the electrodes were properly placed in the vCA1 other than the original targeting coordinates (e.g., post-hoc histology).

Minor points

7. Since only the merged microscopic images were shown in Figure 1H-1J, it would be better to show single-color images of each in WT and Shank3-KO mice (In addition, if possible, magnified images at the cellular level.). In particular, the staining of VGLUT1 and 3R-Tau is virtually unobservable, making it impossible to interpret data on differential expression between Shank3 and WT.

8. Why is the control virus used by the authors AAV-CaMKIIa-GFP (instead of AAV-CaMKIIa-mCherry)? Since pAAV is a DNA plasmid, I think AAV is the correct term. Furthermore, the serotype should be clearly stated. (l.219, 293)

9. I could not find the definition of “immobility time” (ll. 334-335).

10. Please describe the details of “video recording” (l. 585) such as equipment and recording settings (e.g. frame rate).

11. It seems clearer to cite Tao et al., 2022 alongside Rao et al., 2019 in “both the CA2 (Oliva et al., 2020) and ventral CA1 (vCA1) (Rao et al., 2019)” (l.91), as the ripple activity study within the hippocampus that is most relevant to this study.

12. I think the comparison between D and E in Figure 2 is a very important point; why not use three-way ANOVA?

Response to Referees

The reviewers' comments are pasted verbatim below in bold font. They are followed by our responses in non-bold text. Changes to the manuscript are indicated in red font.

Reviewer #1:

This impressive manuscript investigates hippocampal neural circuitry underlying social discrimination deficits in Shank3B knockout mice. The authors conducted elegant chemogenetics studies which revealed that activating excitatory neurons in the CA2 region, and activating CA2 afferents to vCA1, restored social discrimination two weeks later in Shank3B knockout mice. The social recognition improvement was accompanied by electrophysiological increases in vCA1 theta power in the Shank3B knockouts. Another intriguing finding was a lower intensity of 3R-Tau+ mossy fibers in the CA2 of Shank3B mutant mice.

Methods for the three trial direct social interaction test for recognition memory, and for DREADD administration, are generally well described. Appropriate statistical analyses were used for the behavioral data. Sex differences were not detected. A control comparison using object location memory confirmed that non-social recognition memory was intact in both genotypes. Elevated plus-maze experiments confirmed similar anxiety-related behaviors in WT and Shank3B knockouts, ruling out general avoidance as a potential artifact. The authors are to be complimented on their thorough experimental design, and their insightful interpretations of the many interesting results.

We appreciate the reviewer's description of our manuscript as "impressive" with a "thorough experimental design" and "insightful interpretations of the many interesting results."

Minor additions are needed to further improve the manuscript:

1. In the Methods section, first paragraph describing Animals, please state explicitly that the subject mice were null mutant knockouts (Shank3B -/-), or if they were heterozygotes (Shank3B +/-).

We have done this in the revised methods section of the manuscript.

2. Please provide considerably more details about the methods used to score the "direct sniffing of the stimulus mouse's anogenital region or body, following, or allogrooming that was initiated by the test mouse." (page 18). Include information about exactly how the rater was kept blind to genotype and treatment condition.

We have added the requested information to the methods section.

3. Please describe the object(s) used for the object location memory test.

We have added the requested information to the methods section.

4. As described in paragraph 4 of the Introduction, many aspects of social behavior are impaired in people with autism spectrum disorder. However, usually there is not a problem in remembering specific faces if the subject has been able to look directly at the new person's face sufficiently. Premising a manuscript about social discrimination on autism symptoms is an overreach. The first two sentences state: "Social memory is an important ability that gives rise to adaptive social interactions. Social memory dysfunction is a characteristic of several neuropsychiatric

disorders, including autism spectrum disorder (ASD), schizophrenia, and major depressive disorder (Williams et al., 2005; Porcelli et al., 2019; Schafer and Schiller, 2019).” This statement will require considerably more descriptions and relevant references to convince the reader that there is evidence for a serious social recognition memory dysfunction in people with autism.

In response to this comment, we have added several references showing that humans with ASD have difficulty recognizing faces and voices, especially when there is a time delay between presentations. One of the cited articles is a review by Weigelt, Koldewyn, and Kanwisher (2012) that identifies and discusses the results of 6 published articles (representing 7 experiments) that show positive evidence for impaired recognition of familiar faces. Many of these articles also show no evidence for impaired recognition of objects in people with ASD, suggesting that the deficit is specific to social stimuli. In the revised manuscript, we cite some of this literature in the introduction. Since *SHANK3* is a risk gene identified in ASD GWAS studies, and is a known causal factor in Phelan-McDermid syndrome, a neurodevelopmental condition that is typically comorbid with ASD symptoms, including social recognition deficits, we do not feel that this rationale is an “overreach”. However, we have added a sentence at the end of the discussion to make it clear that it remains unknown whether our results are relevant to people with ASD. The reviewer also pointed out that social recognition deficits may be related to people with ASD paying less attention to social stimuli. We mention this in our discussion in the context of Phelan-McDermid syndrome, and feel that it may be relevant to our *Shank3B* mouse findings, because of the low novel investigation times we observed.

Reviewer #2:

The study by Cope et al tests the hypothesis that CA2 neuron dysfunction is (at least in part) responsible for social discrimination phenotypes seen in Shank3b KO mice. This work builds upon several previous studies in the last 5-10 years demonstrating that acute or chronic silencing of CA2 neurons in male mice disrupts social novelty discrimination. The dCA2-vCA1 is a well described obligatory circuit for the expression of social recognition memory (PMID: 30301899 PMID: 35811321). There are several extrahippocampal CA2 afferents that have been shown to influence social recognition memory (SuM, PVN, MSDB, LEC), but not all are necessary for the expression social memory. There have been a few studies identifying CA2 specific neuron impairments as a substrate for social memory dysfunction in mouse models (22q11DS, NLG3 KO, sorCS2 KO), but the field is relatively new, and the current body of work adds to this growing and exciting field. There are also several papers investigating social deficits and relevant circuits in Shank3b KO mice (PMID: 32024781, PMC5472997, PMID: 31061091) although none, to my knowledge, that investigate area CA2. Thus, the topic addressed in Cope et al is novel, timely and of broad interest.

We appreciate the reviewer describing our manuscript as “novel, timely and of broad interest”.

Using primarily chemogenetic activation approaches the manuscript describes the main finding that activation of CA2 neurons or terminal-specific activation of the dCA2-vCA1 circuit restores social approach and social discrimination behaviors in Shank3b KO mouse. The authors provide histological and local field potential data to exclude the possibility that social behavior deficits in Shank2b mice are due to disruption in (a subset of) anatomical inputs to CA2 or to differences in high frequency oscillations (low gamma and SWR), respectively. However, the authors do not uncover an underlying mechanism (aside from correlative theta activity) supporting how CA2 neuron activity causally promotes social behavior in Shank2b mice, which limits the excitement over the otherwise well designed and executed study. It is not clear whether there is a CA2 neuron deficit in Shank3b mice that contributes to the social phenotype, but the data clearly show that CA2

activation can promote or improve social behavioral phenotypes in these mice. Differentiating between these two points is an important distinction that warrants more discussion.

We agree that our findings do not uncover an underlying mechanism for social discrimination dysfunction although we have found a way to reverse this deficit in adulthood. As recommended by the reviewer, we include more discussion on the distinction between these two types of findings.

Specific comments

The experiments appear to be done with technical rigor and in a sufficient number of male and female mice to support the main funding of the paper. The use of both sexes is welcomed as there is a paucity of social behavioral data after CA2 manipulations in female mice.

Fig 1 B it is not clear what the asterisk next the wt is indicating—the overall effect of genotype x trial? In general, the positioning of asterisks for significance was a bit confusing throughout the figures.

In response to this comment, we have changed the positioning of the asterisks and made it clear what they refer to in the figure legends.

Fig 1 H-J: red/green merged images are not color blind friendly. Please edit the color scheme and/or show the individual channels separately so the reader can see the quantified staining in question. Representative images per genotype are also welcomed, especially for J (3R-Tau) where there is a significant decrease in MF staining.

In response to this comment, we have changed the image colors so that they can be visualized by color blind readers. We also show the individual channels and representative images per genotype for all of the afferent labels examined (VGLUT1, ZnT3, VGLUT2, VACht, 3R-Tau.). These images are shown in Figures 1, S2, S3, and S4.

I encourage the authors to avoid direct comparisons of sociability and interaction times between WT and KO mice, such as on page 5 results mid paragraph, “they [KO] also had lower novel interactions times as compared to WT, perhaps indicating reduced sociability”. The authors who established the initial version of the task that is slightly varied here (3-chamber sociability and social novelty task) states in PMC4904775 that “Sociability is a yes-or-no phenotype, obtained by comparing two sides within each group. Sociability is NOT a graded parameter for quantitatively comparing chamber times across groups.” This is underscored by later results (Sup Fig 4) demonstrating that overall mobility/locomotion/ exploration is different between KO and WT, making the comparison difficult to interpret.

We agree with this reviewer’s comments and have replaced the term “sociability” with investigation of novel social stimuli.

If the authors wish to report cross group comparisons on direct measures (i.e., exploration time), please report the F and p stats for the main interaction effect of genotype x trial within the text to justify it (provided in sup table 1). Alternatively, basing conclusions on the statistical result described in Fig 1C is more appropriate, while also acknowledging the limitations of the hypomobility phenotype in Shank3b KO mice as previously shown (PMC5472997).

We now base the conclusions on the statistical result described in Figure 1C and acknowledge the limitations of the hypomobility phenotype in *Shank3B* KO mice.

While the rationale for disrupted afferents to CA2 as a potential locus for CA2 dysfunction resulting in social deficits is logical, the intrahippocampal inputs listed (EC/DG/CA3) are not directly linked to social memory. EC to DG involvement in social memory (Leung et al 2018) has been reinterpreted based on the findings reported in Lopez-Rojas et al and Dang et al (<https://doi.org/10.1007/s12264-021-00813-6>) that suggest the impairment in initially attributed to EC-DG was due to unintended inhibition of EC-CA2 fibers of passage. Chiang et al implicated ventral CA3 to social memory, and to my knowledge ventral CA3 does not project to dorsal CA2. Lopez-Rojas et al also showed that DG circuits (DG-CA3-CA1, DG-CA2-CA1, DG-CA3-CA2-CA1) are dispensable for social memory. I recommend refocusing the rationale without linking the afferents mentioned directly to social memory and highlighting the (lack of) differences in social memory relevant inputs from SuM and MSDB.

We agree with the reviewer's comments and have refocused the rationale for looking at CA2 afferents as suggested.

While I appreciate the fact that if gross anatomical differences in inputs were uncovered it would likely have functional implications, however because few were found, it begs the question as to whether the inputs present in normal numbers are functionally impaired, perhaps in NMDAR trafficking at the synapse. This limitation should be addressed.

We agree with the reviewer's comment that NMDA receptor trafficking at the synapse might be impaired in *Shank3B* KO mice and we now address this possibility in the discussion.

Fig 2B what percentage of PCP4+ cells were DREADD+? Based on the image, it doesn't appear to be 100% and could explain differences across CA2 Gq DREADD studies (i.e., Alexander et al 2018)

In response to this query, we counted the number of PCP4+ cells that were mCherry+ in a subset of brains and found that the percentage was about 51%. Alexander et al (2018) reported over 90% labeling of PCP4+ cells with DREADD virus. Due to the small size of the CA2 and the fact that our study did not use a cre mouse line that would restrict labeling to the CA2, we used a much smaller injection volume of DREADD virus than the Alexander et al (2018) paper (15 nl compared to 500 nL), which likely led to the infection of fewer PCP4+ cells in the current study. This adds support to the reviewer's suggestion that differences in labeling may explain different electrophysiological results in the two studies. In the revised manuscript, we include the new data and discuss this point. However, despite our lack of gamma and SWR effects with DREADD-induced activation, we observed a relevant behavioral change. This suggests that a sufficient number of pyramidal neurons were activated in our studies to have a functional effect.

Fig 3C No mention of the fact that WT DREADD vehicle controls failed to discriminate novel vs familiar.

In the revised manuscript, we now mention this point in the results section, which we believe is due to a few mice that showed a paradoxical reverse social preference, for unknown reasons.

Fig 4 were theta/gamma epochs measured during periods of mobility or quiet rest/investigation of social stimulus? There should be larger effect on gamma during periods of mobility. I believe the authors intend to cite Alexander et al 2018 not 2016 as used throughout. Alexander et al recorded from intermediate CA1 with CA2 Gq DREADDs. Can the authors present representative spectrograms with mobility data in addition to the power measurements? Interestingly, in a previous EEG study, an increase in gamma power was detected in Shank3b mice (PMC5472997), although there could be differences based on region.

In response to this comment, we measured low gamma power during periods of mobility and found no differences between *Shank3B* KO and WT. These data are now included in Figure S10. The reasons for the difference between our findings and Alexander et al 2018 may be related to differences in recording region (intermediate CA1 versus ventral CA1) or the earlier mentioned difference in the percentage of PCP4+ cells that were DREADD+ between studies. These possibilities are discussed in the paper and the citation has been corrected.

Reviewer #3:

The authors have successfully treated social memory abnormalities in Shank3-KO mice, a mouse model of autism, by manipulating the activity of specific memory circuits in the hippocampus, which is interesting and highly promising research. Improvements are listed below.

We appreciate that the reviewer finds our work “interesting and highly promising research”

Major Points

The authors claim to manipulate CA2 neuronal activity by microinjecting AAV specifically into the CA2 region (Figure 2B), but CA2 is a very small region, so it is extremely difficult to control without using CA2-Cre mice. The extent to which DREDD expression was observed in CA3/CA1 and the extent to which DREDD was not expressed in CA2 neurons should be examined. Presumably, the authors have confirmed the expression of AAV in the CA2 region in all mice that have completed behavioral testing, but can the data show this?

In response to this comment, we now include images showing DREDD infection in the CA2 (Figure S6). We confirmed expression of AAV in the CA2 in all mice included in the study. We also found very little infection outside of the CA2 and mention this in the results. Two mice were removed due to missed injections – 1 from the cannula infusion study and 1 from the electrophysiology study. These data sets were reanalyzed to account for the removed mice. These changes did not alter the results. The removed mice are described in the revised manuscript.

Smith et al., (Mol. Psy. 21:1137, 2016) show that social memory lasts a week when CA2 neurons are hyperactivated. Therefore, in the case of the experiment in Figure 2, there should be more than one week between Behavior 1 and Behavior 2. If not, it is necessary to distinguish between the group that received CNO at Behavior 1 and Behavior 2, and carefully discuss it.

In response to this comment, we analyzed our *Shank3B* KO DREDD data by separating mice that received CNO first and comparing their behavior to mice that received vehicle first. We found that both groups showed evidence of ability to discriminate between novel and familiar mice. Similar data between the groups (CNO first vs vehicle first) suggest that testing order did not substantially influence the effect. The separated data are now provided in Figure S7.

The authors make a very generalized claim saying that “SWRs are similar between WT and Shank3B KO mice” (l. 325), even though only the first minute of Trial 2 was included in the analysis, which appears to be arbitrary criteria. More comprehensive data (e.g., SWR numbers, amplitude, duration, frequency [Hz], travel distance, immobility time minute by minute during the social interaction, as well as the baseline trial) would be helpful in understanding if presented.

In response to this concern, we carried out multiple additional analyses on our SWR electrophysiological data, examining numbers, amplitude, duration, and spacing interval between SWRs during each minute of the baseline and familiar conspecific trials. We found no substantial differences on any of these measures

between *Shank3B* KO vs WT or between *Shank3B* KO DREADD vehicle vs CNO. These data are now reported in Figures S12-S15. We detected a few significant differences between WT DREADD vehicle vs CNO (SWR numbers at minute 2 of the familiar conspecific trial: CNO is greater than vehicle, $p=0.00140$; SWR interval time at minute 2 of the familiar conspecific trial: CNO is lower than vehicle, $p=0.0050$), but these effects were not seen during any other minutes of the trial, nor did they correspond to any of our behavioral findings.

Additionally, in the Discussion, the authors carefully mention the limitations of their analysis, while the abstract uses the very vague sentence; "we observed no differences in these measures between WT and Shank3B KO mice in the vCA1 in response to social stimuli". This section is extremely misleading and should be improved.

In response to this concern, we have modified the phrasing in the abstract to address the fact that the lack of differences we observed was on the measures we examined and does not rule out the possibility that other, not yet examined, electrophysiological measures may differ.

SWR frequency (l. 337) should be expressed in more precise units, such as the number of SWRs per second during the immobile time. I am wondering why the authors excluded this most striking phenotype (~50% decrease of SWR occurrence in KO mice!) from their main claims. Is it conceivable that the decrease of SWR is attributable to the diminished SWR amplitude as a result of the exclusion of the SWRs below the threshold of 3SD?

The reviewer questioned why we did not feel that the lower frequency of SWRs in *Shank3B* KO compared to WT was an important electrophysiological finding of the manuscript. We included the finding in the text because it is a standard way that researchers calculate SWR frequency, but do not think it reflects a real difference since the measure was calculated with a denominator of immobility time (because SWRs occur during periods of sleep and quiet wakefulness). Since there is a large difference in immobility time between genotypes, this measure produces an artificially lower frequency for the group with greater immobility (*Shank3B* KOs). When we calculated absolute SWR number, the difference between genotypes disappeared suggesting that within a given time period, *Shank3B* KO and WT mice have a similar number of SWRs. This reasoning is included in the paper and in response to the reviewer's comments, we now display SWR number during each minute of baseline (3 minutes) and familiar mouse (5 minutes) trials. We also analyzed whether there were differences in the interval of time between SWRs in both genotypes and found no differences. Because we applied a standard cut-off for amplitude in our definition of a SWR, it remains possible that we missed some "SWR-like" events in *Shank3B* KOs if they were lower than WTs. However, this seems unlikely given the similarities in so many SWR measures between genotypes, which suggest that social discrimination differences are not related to the SWR measures we made. We have included this possibility in the revised manuscript.

It is not clear how "SWR numbers, amplitude, and duration were ... normalized by dividing by the baseline trial" (l. 637-639). Does this mean that SWRs during Trial 2 were almost three times larger than those recorded during the baseline trial in amplitude (as shown in Figure 4G; which would hardly be the case)?

The reviewer is correct that the SWR normalization procedure we used was not appropriate for amplitude and duration, because the baseline trial contained 3 minutes of data while we were normalizing to only 1 minute. We thank the reviewer for noticing this error and have corrected this in the revised manuscript, including in Figures 4, S11.

The authors implanted a custom-made 3 wire electrode array and analyzed each electrode as separate samples, which should be statistically treated as a single sample.

We agree with the reviewer's comment that treating each electrode as a separate sample is incorrect and we have changed our statistical analysis of the electrophysiological data as a result. However, because electrophysiological measurements were taken from multiple electrodes within each mouse, we chose to use a linear mixed-effects ANOVA, which considers not only the number of mice but the number of samples per mouse. The level of the measurement was explained by drug, virus, genotype, the three two-way interactions, the three-way interaction, and a random effect of mouse. Tukey post hoc comparisons were used to follow up any significant main effects or interactions. The new analyses are described in the methods section and the results are presented in Figures 4, S10-S15 and the statistics tables.

Also, it should be stated how the authors confirmed that the tips of the electrodes were properly placed in the vCA1 other than the original targeting coordinates (e.g., post-hoc histology).

All mice were verified for electrode placement in the electrophysiology experiment and for cannula placement in the vCA1 infusion experiment by being counterstained with Hoechst 33342.

Minor points

Since only the merged microscopic images were shown in Figure 1H-1J, it would be better to show single-color images of each in WT and Shank3-KO mice (In addition, if possible, magnified images at the cellular level.). In particular, the staining of VGLUT1 and 3R-Tau is virtually unobservable, making it impossible to interpret data on differential expression between Shank3 and WT.

As mentioned above, we now include single color images in WT and KO mice for all afferent labels (Figures S2, S4). We also show magnified images of the two markers that are difficult to visualize, VGLUT1 and 3R-Tau in Figure S3.

Why is the control virus used by the authors AAV-CaMKIIa-GFP (instead of AAV-CaMKIIa-mCherry)? Since pAAV is a DNA plasmid, I think AAV is the correct term. Furthermore, the serotype should be clearly stated. (l.219, 293)

We used a control virus that expressed GFP since it had a matched serotype and promoter. The control virus produced comparable infection to our DREADD virus. In response to this comment, we have clearly stated the serotype and changed pAAV to AAV.

I could not find the definition of “immobility time” (l. 334-335).

In response to this comment, we define immobility time in the revised manuscript.

Please describe the details of “video recording” (l. 585) such as equipment and recording settings (e.g. frame rate).

In response to this comment, we now include details about video recording, including equipment and frame rate.

It seems clearer to cite Tao et al., 2022 alongside Rao et al., 2019 in “both the CA2 (Oliva et al., 2020) and ventral CA1 (vCA1) (Rao et al., 2019)” (l.91), as the ripple activity study within the hippocampus that is most relevant to this study.

In response to this comment, we cite Tao et al., 2022 alongside Rao et al.

I think the comparison between D and E in Figure 2 is a very important point; why not use three-way ANOVA?

In response to this suggestion, we have examined whether the difference score (difference in time investigating novel and familiar mice) changed within subject in response to drug treatment. Using a three-way ANOVA and Bonferroni post hoc comparisons, we found that KO + DREADD have higher difference scores following CNO treatment compared to VEH treatment ($p= 0.0061$), while all other groups showed no change in difference scores between VEH and CNO treatment. We now display this comparison (Figure S8) and report the detailed statistics (see Table S2).

REVIEWER COMMENTS

Reviewer #1 (Remarks to the Author):

The authors have successfully addressed the previous concerns.

Comments on Response to Reviewer #2 (Remarks to the Author):

I have examined all the questions from reviewer #2 and the author's answers to them.

Frankly, it seems to me that the authors need to add some explanation in the manuscript and experiments to address the main concern of #2.

(In addition, it is very difficult to know if the corrections have been made or not, since it is not clearly indicated in the rebuttal letter where the author corrected the sentences in the manuscript...)

For example,

Reviewer#2's Question: Using primarily chemogenetic activation approaches the manuscript describes the main finding that activation of CA2 neurons or terminal-specific activation of the dCA2-vCA1 circuit restores social approach and social discrimination behaviors in Shank3b KO mouse. The authors provide histological and local field potential data to exclude the possibility that social behavior deficits in Shank2b mice are due to disruption in (a subset of) anatomical inputs to CA2 or to differences in high frequency oscillations (low gamma and SWR), respectively. However, the authors do not uncover an underlying mechanism (aside from correlative theta activity) supporting how CA2 neuron activity causally promotes social behavior in Shank2b mice, which limits the excitement over the otherwise well designed and executed study. It is not clear whether there is a CA2 neuron deficit in Shank3b mice that contributes to the social phenotype, but the data clearly show that CA2 activation can promote or improve social behavioral phenotypes in these mice. Differentiating between these two points is an important distinction that warrants more discussion.

Authors's answer: We agree that our findings do not uncover an underlying mechanism for social discrimination dysfunction although we have found a way to reverse this deficit in adulthood. As recommended by the reviewer, we include more discussion on the distinction between these two types of findings.

My opinion:

This question was the main question of Reviewer #2. Perhaps Reviewer #2 was expecting a careful discussion of the mechanism by which artificially induced excitation of the CA2 neurons resulted in the recovery of social memory behavior (based on Shank3 expression and function in the hippocampal microcircuits), but the authors only added the following sentence in Discussion, "Thus, although our

experiments did not uncover a mechanism underlying the social deficit in Shank3B KO mice, we have identified a manipulation that can restore this behavior to WT levels."

Reviewer#2's Question: Fig 3C No mention of the fact that WT DREADD vehicle controls failed to discriminate novel vs familiar.

Authors's answer: In the revised manuscript, we now mention this point in the results section, which we believe is due to a few mice that showed a paradoxical reverse social preference, for unknown reasons.

My opinion: This is a very important experiment that supports the main claim of this paper. It is difficult to understand why the authors do not examine in depth experimentally the unknown results of the control experimental group.

Reviewer#2's Question: While the rationale for disrupted afferents to CA2 as a potential locus for CA2 dysfunction resulting in social deficits is logical, the intrahippocampal inputs listed (EC/DG/CA3) are not directly linked to social memory. EC to DG involvement in social memory (Leung et al 2018) has been reinterpreted based on the findings reported in Lopez-Rojas et al and Dang et al (<https://doi.org/10.1007/s12264-021-00813-6>) that suggest the impairment in initially attributed to EC-DG was due to unintended inhibition of EC-CA2 fibers of passage. Chiang et al implicated ventral CA3 to social memory, and to my knowledge ventral CA3 does not project to dorsal CA2. Lopez-Rojas et al also showed that DG circuits (DG-CA3-CA1, DG-CA2-CA1, DG-CA3-CA2-CA1) are dispensable for social memory. I recommend refocusing the rationale without linking the afferents mentioned directly to social memory and highlighting the (lack of) differences in social memory relevant inputs from SuM and MSDB.

Authors's answer: We agree with the reviewer's comments and have refocused the rationale for looking at CA2 afferents as suggested.

My opinion: Again, authors almost ignored the reviewer#2's suggestion. As far as I checked the entire manuscript, no particular discussion has been revised on how the present findings are to be interpreted in an integrated manner with previous findings on microcircuits within the hippocampus.

Reviewer #3 (Remarks to the Author):

The authors have clearly addressed all of my concerns through reanalysis and additional experiments. The paper is considered to be complete and of high scientific value.

Response to reviewers

Reviewers' comments are indicated in bold black font. Our previous responses are indicated in plain black font. Our responses to the most recent set of comments are indicated in plain blue font.

Reviewer #1

The authors have successfully addressed the previous concerns.
We appreciate that the reviewer is satisfied with our revised manuscript.

Comments on Response to Reviewer #2:

I have examined all the questions from reviewer #2 and the author's answers to them. Frankly, it seems to me that the authors need to add some explanation in the manuscript and experiments to address the main concern of #2.
(In addition, it is very difficult to know if the corrections have been made or not, since it is not clearly indicated in the rebuttal letter where the author corrected the sentences in the manuscript...)

We apologize for not indicating the location of the corrected sentences in the manuscript. In this version, both new and old text that is relevant to the reviewer's concerns has been placed in blue font. In addition, the page numbers where the relevant text is included are mentioned in response to this reviewer's comments.

For example,

Reviewer#2's Question: Using primarily chemogenetic activation approaches the manuscript describes the main finding that activation of CA2 neurons or terminal-specific activation of the dCA2-vCA1 circuit restores social approach and social discrimination behaviors in Shank3b KO mouse. The authors provide histological and local field potential data to exclude the possibility that social behavior deficits in Shank2b mice are due to disruption in (a subset of) anatomical inputs to CA2 or to differences in high frequency oscillations (low gamma and SWR), respectively. However, the authors do not uncover an underlying mechanism (aside from correlative theta activity) supporting how CA2 neuron activity causally promotes social behavior in Shank2b mice, which limits the excitement over the otherwise well designed and executed study. It is not clear whether there is a CA2 neuron deficit in Shank3b mice that contributes to the social phenotype, but the data clearly show that CA2 activation can promote or improve social behavioral phenotypes in these mice. Differentiating between these two points is an important distinction that warrants more discussion.

Authors's answer: We agree that our findings do not uncover an underlying mechanism for social discrimination dysfunction although we have found a way to reverse this deficit in adulthood. As recommended by the reviewer, we include more discussion on the distinction between these two types of findings.

My opinion:

This question was the main question of Reviewer #2. Perhaps Reviewer #2 was expecting a careful discussion of the mechanism by which artificially induced excitation of the CA2 neurons resulted in the recovery of social memory behavior (based on Shank3 expression and function in the hippocampal microcircuits), but the authors only added the following sentence in Discussion, "Thus, although our experiments did not uncover a mechanism underlying the

social deficit in Shank3B KO mice, we have identified a manipulation that can restore this behavior to WT levels."

In response to this concern, we now include more discussion about a potential mechanism of artificially induced recovery of social memory behavior. This discussion is present on pages 17-18.

Reviewer#2's Question: Fig 3C No mention of the fact that WT DREADD vehicle controls failed to discriminate novel vs familiar.

Authors's answer: In the revised manuscript, we now mention this point in the results section, which we believe is due to a few mice that showed a paradoxical reverse social preference, for unknown reasons.

My opinion: This is a very important experiment that supports the main claim of this paper. It is difficult to understand why the authors do not examine in depth experimentally the unknown results of the control experimental group.

We agree that it is surprising that unlike the other control groups in the experiment shown in Figure 3 (WT control virus + vehicle, WT control virus +CNO, WT DREADD virus + CNO), the WT DREADD+ vehicle control group's overall decrease in investigation time between novel and familiar stimulus presentations did not reach statistical significance.

We and others have found that adult control mice occasionally show a reverse social preference with greater investigation of familiar than novel stimulus mice (Laham et al., 2021; Cope et al., 2022; Wu et al., 2021; Lopez-Rojas et al., 2022). In this particular group, two mice had this behavioral profile. It is important to note that this is not an absence of social discrimination, but a reversal of preference, indicating that there is an intact ability to discriminate novel from familiar social stimuli. We typically do not remove these mice from the analysis, despite the fact that they are biological outliers, because they likely reflect the normal variation in mouse behavior. We rescored the videos from these mice and confirmed our original results. We also did not observe any atypical behaviors with these mice other than their reversed social preference and feel that it is valid to include them in the analysis.

While it would be ideal to conduct an in-depth experimental examination of the unexpected results of this control group, we do not feel that the number of mice required to potentially balance out the chance variation can be justified by the contribution it would offer to the paper. We feel that the consistency of the controls, particularly in the other control groups and across experiments, and among all but two control mice in the group in question, and the robustness of the paper's points are sufficiently supported despite the reversed discrimination of these two mice.

It should be noted that less stringent statistical tests that do not correct for multiple comparisons reveal a statistical difference between novel and familiar stimulus exposure in this group, but we have elected to use more stringent tests throughout the entire paper, in addition to providing all of the data so that the reader can evaluate it in its entirety. In the revised manuscript we now describe the results of this unexpected control group in more detail in the results section on pages 9-10.

Reviewer#2's Question: While the rationale for disrupted afferents to CA2 as a potential locus for CA2 dysfunction resulting in social deficits is logical, the intrahippocampal inputs listed (EC/DG/CA3) are not directly linked to social memory. EC to DG involvement in social memory (Leung et al 2018) has been reinterpreted based on the findings reported in Lopez-Rojas et al and Dang et al (<https://doi.org/10.1007/s12264-021-00813-6>) that suggest the impairment in initially attributed to EC-DG was due to unintended inhibition of EC-CA2 fibers of passage. Chiang et al implicated ventral CA3 to social memory, and to my knowledge ventral CA3 does not project to dorsal CA2. Lopez-Rojas et al also showed that DG circuits (DG-CA3-CA1, DG-CA2-CA1, DG-CA3-CA2-CA1) are dispensable for social memory. I recommend refocusing the rationale without linking the afferents mentioned directly to social memory and highlighting the (lack of) differences in social memory relevant inputs from SuM and MSDB.

Authors's answer: We agree with the reviewer's comments and have refocused the rationale for looking at CA2 afferents as suggested.

My opinion: Again, authors almost ignored the reviewer#2's suggestion. As far as I checked the entire manuscript, no particular discussion has been revised on how the present findings are to be interpreted in an integrated manner with previous findings on microcircuits within the hippocampus.

We apologize if it appeared that we ignored this suggestion. It was not our intention to do so. In the new version of the manuscript, we more specifically mention concerns of the reviewer and focus more on the lack of differences in SuM and MSDB inputs. We have indicated all of the relevant text in blue font – this text is present in the results section (pages 6-8) and in the discussion section (pages 13,14,15).

Reviewer #3 (Remarks to the Author):

The authors have clearly addressed all of my concerns through reanalysis and additional experiments. The paper is considered to be complete and of high scientific value.

We appreciate that reviewer #3 was satisfied with our revision and considers the paper to be of “high scientific value.”